# CMOS Fixed Pattern Noise Removal Based on Low Rank Sparse Variational Method

**Tao Zhang [1,2,3,*], Xinyang Li [1], Jianfeng Li [2] and Zhi Xu [3]**

[1] Key Laboratory of Adaptive Optics, Institute of Optics and Electronics, Chinese Academy of Sciences, Chengdu 610209, China; xyli@ioe.ac.cn

[2] School of Optoelectronic Information, University of Electronic Science and Technology, Chengdu 611731, China; lijianfeng@uestc.edu.cn

[3] Astronomical Technology Laboratory, Yunnan Observatory, Chinese Academy of Sciences, Kunming 650216, China; xuzhi@ynao.ac.cn

\* Correspondence: ztao@ynao.ac.cn

**Abstract:** Fixed pattern noise (FPN) has always been an important factor affecting the imaging quality of CMOS image sensor (CIS). However, the current scene-based FPN removal methods mostly focus on the image itself, and seldom consider the structure information of the FPN, resulting in various undesirable noise removal effects. This paper presents a scene-based FPN correction method: the low rank sparse variational method (LRSUTV). It combines not only the continuity of the image itself, but also the structural and statistical characteristics of the stripes. At the same time, the low frequency information of the image is combined to achieve adaptive adjustment of some parameters, which simplifies the process of parameter adjustment, to a certain extent. With the help of adaptive parameter adjustment strategy, LRSUTV shows good performance under different intensity of stripe noise, and has high robustness.

**Keywords:** FPN; low rank; sparse; total variation; anisotropy; characteristic

---

## 1. Introduced

Compared with CMOS Image Sensor (CIS), CCD has the characteristics of high quantum efficiency, high sensitivity, low dark current, good consistency and low noise. However, in recent years, with the development of large-scale integrated circuit technology, the photoelectric characteristics of CIS have been greatly improved. In particular, sCMOS sensor is a composite technology which combines the advantages of CCD and CMOS. It also has the features of high quantum efficiency, high sensitivity and low dark current [1,2]. However, CIS still lags behind CCD in the consistency of photoelectric response. This is mainly due to the apparent fixed pattern noise (FPN) in CIS relative to the CCD. However, CIS is also favored by many industries, due to its outstanding advantages in high acquisition rate and low cost. Technically speaking, the appearance of FPN is mainly caused by the structure of CIS. In order to reduce the readout noise and improve the signal-to-noise ratio, most CIS uses active pixel structures such as three transistors (3T), four transistors (4T), five transistors (5T). In each pixel, the electrons generated by the photoelectric response also need to pass the pixel amplifier and the column amplifier to reach the ADC and the digital processing unit finally. Due to the mismatch of pixel amplifier and column amplifier, the photoelectric characteristics of pixels and columns are inconsistent, which leads to the appearance of FPN. The reason for FPN generation is analyzed in detail, with the 3T structure in Figure 1 as an example.

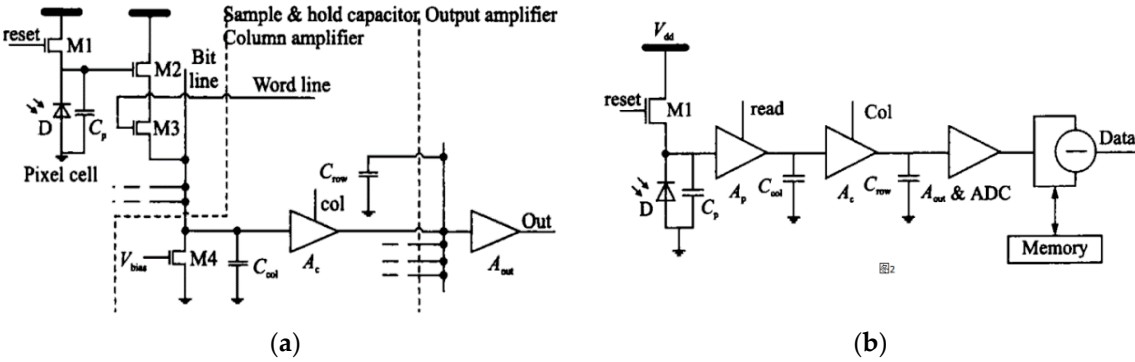

**Figure 1.** A 3T circuit structure of CMOS Image Sensor (CIS) single pixel. (**a**) A 3T active pixel structure. (**b**) A 3T structure equivalent circuit.

Due to the difference in the threshold voltage of different transistors, different output results can be obtained even under the same light conditions. For Figure 1b analysis, this is due to the inconsistency of $A_p$ magnification among different pixels. The noise caused by this $A_p$ inconsistency is called pixel fixed pattern noise (PFPN). It appears as a snowflake patch on the image. In general, PFPN can be suppressed to a great extent by using the related double sampling technique (CDS). The M4 load transistor, AC amplifier and Aout amplifier are output structures shared by each column pixel. Due to the mismatch between the bias voltage of each column amplifier, a noise called column fixed pattern noise (CFPN) is produced, which is shown as vertical stripes on the image. In summary, the fixed pattern noise can be made up of two parts: FPN = CFPN + PFPN, where FPN is the total fixed pattern noise, PFPN is pixel fixed pattern noise, and CFPN is column fixed pattern noise. The image with FPN noise is similar to Figure 2. To suppress FPN, CDS technology is generally used, but it can only eliminate PFPN component in FPN. CFPN caused by output stage amplifier $A_{out}$ cannot be effectively eliminated. For specific analysis, please refer to the literature [3,4]. After CDS, the output voltage can be described by Formula (1):

$$
\begin{aligned}
M &= G \cdot A_p A_C A_{out} \cdot \eta \cdot \frac{P \cdot t}{h \cdot v} + G \cdot A_{out} \Delta V_{A_{out}} \\
&= G \cdot A_p A_C A_{out} \cdot N + G \cdot A_{out} \Delta V_{A_{out}} \\
&= k \cdot N + b
\end{aligned}
\tag{1}
$$

where $G$ is the analog-to-digital conversion coefficient, unit: $DN/v$, $A_p$ is the equivalent comprehensive amplification factor composed of M2, M3 and M4, unit: $v/e$-, $A_c$ and $A_{out}$ are the amplification factors of buffer operation amplifier and output operation amplifier, respectively and $\Delta V_{A_{out}}$ is the bias voltage of output amplifier. $P$ is the incident light power, $t$ is the exposure time, $h$ is the Planck constant, $V$ is the incident light wavelength, $\eta$ is the CIS quantum efficiency. $N$ is the number of incident electrons. $k = G A_p A_C A_{out}$, $b = A_{out} \Delta V_{A_{out}}$.

Generally speaking, there are two kinds of noises in CIS images: 1. random noise—this kind of noise is generally caused by many factors [5], such as thermal noise, Poisson noise, flicker noise, shot noise, etc; 2. fixed pattern noise—as analyzed above, this type of noise is generally caused by the mismatch between pixels. It has the feature that the noise remains constant between frames under constant working conditions and within a certain period of time. FPN noise mainly presents regular stripes, and the human eye is very sensitive to the regular stripes, so the impact of fixed pattern noise is far greater than random noise [6]. For the processing of random noise, the method of multi frame stack and average is generally used, which can effectively reduce the fluctuation amplitude of random noise after multi frame stack and average. Under certain conditions, the fixed pattern noise is almost stable between different frames. Therefore, the intensity and shape of the noise remain the same after multi frame stacking and averaging. The conclusion is that the Inhomogeneity of CIS is mainly caused by FPN. After CDS processing, the Inhomogeneity is mainly caused by CFPN.

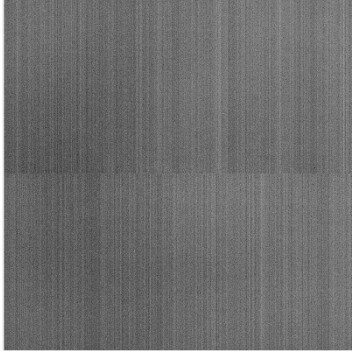

**Figure 2.** Images taken by CIS with fixed pattern noise (FPN) noise (including two-ended output structure).

## 2. Existing Methods

In order to obtain better CIS image quality and reduce the impact of FPN on image quality, many scholars have carried out many studies in this field in recent years. In summary, the FPN removal methods of CIS can be summarized into two categories: 1. The calibration-based method; 2. the scene-based method. As shown in Figure 3.

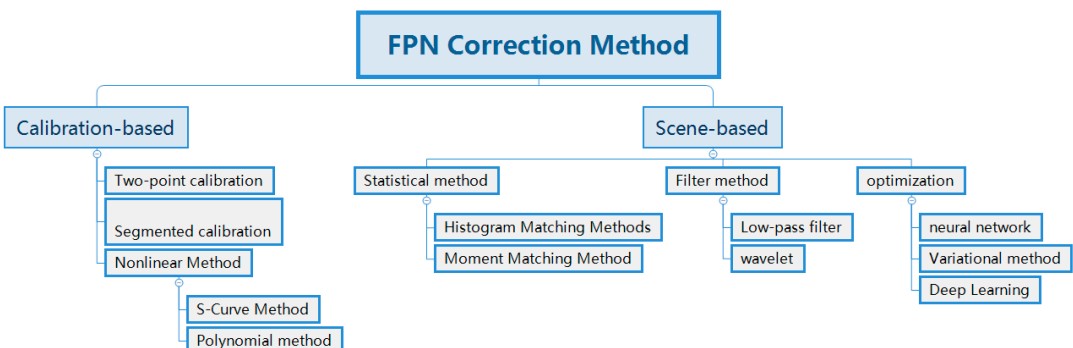

**Figure 3.** Summary of FPN correction methods.

### 2.1. Calibration-Based Method

The main representative methods based on calibration are as follows: the two-point method [7–9], the subsection correction method [10,11], the S-curve method [12,13], the polynomial fitting method [14].

1.  Two-point correction method

The basic idea of two-point correction method is to assume that the photoelectric response characteristic of each pixel is a stable linear relationship, which can be expressed by $M = k \cdot n + B$, where $k$ is the slope of photoelectric response curve, $B$ is the offset, $n$ is the number of incident photons or the energy of incident light. The correction of nonuniformity can be completed by the following steps: (1) calculate the slope correction coefficient g of each pixel; (2) calculate the offset correction coefficient O of each pixel; (3) under the effect of Formula $M = m \cdot G + O$, correct the slope $k$ of each pixel to $K$, and the offset $b$ to $B$, as shown in Figure 4. $K$ is the average of all pixel slopes, and $B$ is the average of all pixel offsets.

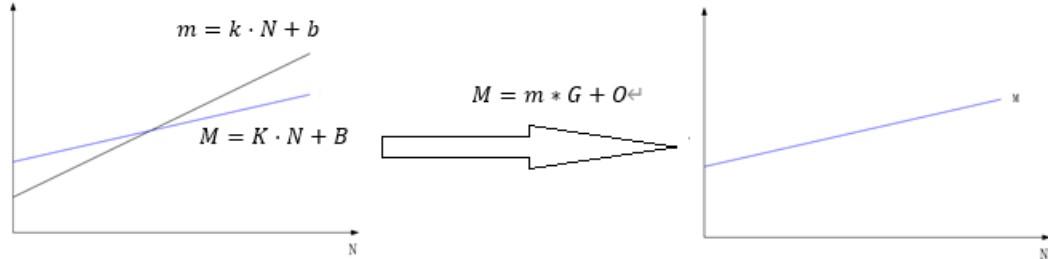

**Figure 4.** Correction process diagram of two-point method (x coordinate: incident light energy y coordinate: gray value).

The premise for the two-point method to be feasible is that the photoelectric response characteristics of each pixel are stable and linear. This is because b in $m = k \cdot N + b$ contains CFPN, as shown in Formula 1. CFPN is due to the bias voltage mismatch of the last op amp, and its mismatch voltage will drift to a certain extent with the working environment and working time [15,16]. Figure 5 shows the whole process of gradual failure of the correction factor as b drifts. First, after 15 min of CIS operation, the corrected parameters $G$ and $O$ are calculated immediately. Next, an image is collected and nonuniformity correction is performed using the currently obtained $G$ and $O$ values. A very good correction result can be obtained at this point, as shown in Figure 5a. After working for CIS for 1 h, the images were collected again, and the original $G$ and $O$ were used for non-uniformity correction. The expected uniform correction results are not obtained at this time, and there is a significant residual of CFPN noise, as shown in Figure 5b. Similarly, CIS works for 3 h, and then collects the image again, and uses the original correction parameters $G$ and $O$ to correct the nonuniformity. The final result is shown in Figure 5c. From the above experiments, it can be very obvious that the calibration-based method can achieve very effective correction in the short term. As time goes by, the original correction parameters become invalid gradually. Therefore, in order to get good correction effect, the calibration parameters $G$ and $O$ need to be calibrated periodically.

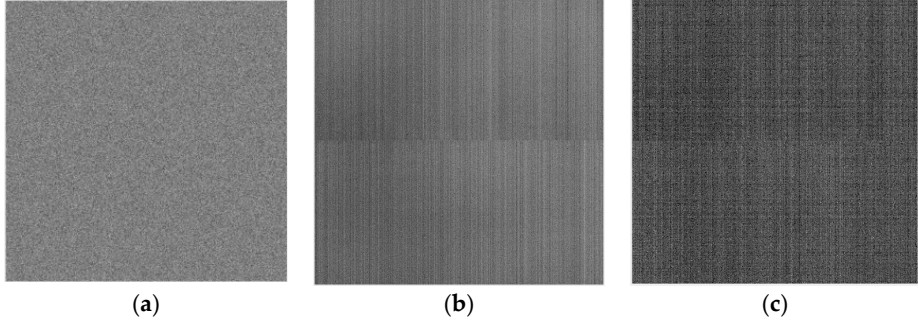

|     |     |     |
| :-: | :-: | :-: |
| (**a**) | (**b**) | (**c**) |

**Figure 5.** The effect of flat field correction (**a**) corrected results after CIS working for 15 min; (**a**) corrected results after CIS working for 15 min; (**b**) corrected results after CIS working for 1 h; (**c**) corrected results after CIS working for 3 h.

## 2.  Segmental Correction Method

Because the linearity of the photoelectric response curve of CIS is not ideal, it shows some nonlinearity, as shown in Figure 6. It can be understood that $k$, $b$ change with the energy of the incident light, that is, $K$ (N), $b$ (N) are a function of the incident energy. Unified use of the same set of correction parameters $G$, $O$ will reduce the accuracy of the correction. In order to obtain a higher correction accuracy, piecewise linear fitting can be used. The curve can be approximated as a combination of several linear segments, each of which is corrected by a two-point method. With the drift of CIS parameters, the piecewise correction method will also encounter the problem of gradually invalidating. Therefore, we need to calibrate the parameters regularly and repeatedly.

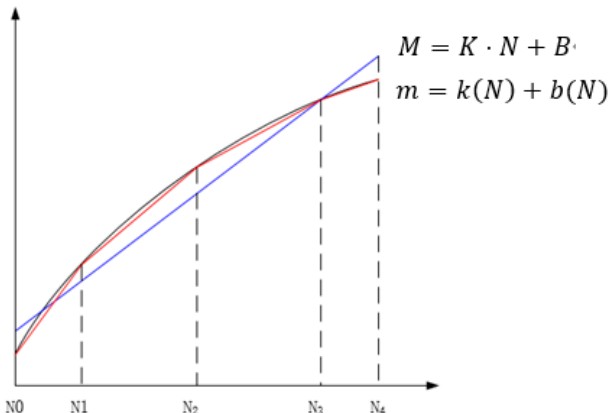

$$M = K \cdot N + B$$
$$m = k(N) + b(N)$$

**Figure 6.** Schematic diagram of sectional correction method (x coordinate: incident light energy; y coordinate: gray value).

3. S-shaped method, polynomial fitting method

To further improve the accuracy of the correction, the photoelectric response curve can be fitted by an S-shaped nonlinear equation or by a polynomial. This type of correction method is computationally intensive and unsuitable for hardware-based real-time nonuniformity correction. Similarly, this kind of method also needs to face the CIS parameter drift problem, and needs to repeat the calibration periodically.

In general, among all calibration-based methods, the two-point method and piecewise linear method are most used in engineering applications, and are very suitable for real-time calibration system, based on field programmable gate array (FPGA). However, these methods need to complete a repeated calibration periodically, which brings great inconvenience to the actual application. Importantly, the correction process is very cumbersome and has strict requirements on the environment and light source.

*2.2. Scene-Based Method*

Because of the operational inconvenience of calibration methods, scene-based noise removal methods have been developed in recent years, which do not require lab calibration. It removes FPN from the image itself. Scene-based methods can be grouped into three categories: 1. filter-based methods [17–19]; 2. statistics-based methods [20,21]; and 3. optimization-based methods [22–30].

1. Filter-based methods

Filter-based methods are widely used [17–19], due to their simplicity and ease of use. The filter-based methods require that the FPN has a certain periodicity in order to have a relatively good effect. In fact, this is difficult to satisfy, and most CIS FPNs are non-periodic. It is difficult to separate the FPN from the image if the filtering method is not used on the basis of periodicity.

2. Statistical methods

The main idea of statistical methods is to modify the distribution of FPN to a reference distribution. For this method to be effective, many statistical similarity assumptions [20,21] must be satisfied. However, at most times, this assumption of similarity is difficult to satisfy, so it is generally difficult to achieve an ideal noise reduction effect.

3. Optimization method

Based on the optimization methods, it has been the most studied methods in recent years. These types of methods can achieve a good denoising effect [22–30]. Essentially, these methods are to solve an underdetermined equation. Most of these methods, which are based on the variational

method, add reasonable prior constraints to complete the transformation of undetermined equations to well-defined equations. The low-rank sparse variational method (LRSUTV), which we will propose next, is also a category of this type of method. At present, some scholars also use the low-rank sparse method to remove stripes. However, most of them remove noise from the image itself, but little attention is paid to the structure of the stripes themselves. A small number of scholars have considered the structural characteristics of the stripes [31–35], but they lack the consideration of the statistical characteristics of the stripes. The stripe removal method we use considers not only the continuity of the image itself, but also the structural and statistical characteristics of the stripes. At the same time, adaptive adjustment measurement of some parameters is added, which simplifies the adjustment process of parameters to a certain extent. Overall, our approach is a more comprehensive approach to noise reduction.

## 3. Motivation for Presenting this Method

Most scene-based correction methods start directly from the observation image to estimate the clear image, and lack some structure information about the strip itself. Then, the structural information of the strip is the key to improve the correction quality. This paper proposes an FPN correction method, based on the details of the image and the stripe itself. By analyzing the structure of CIS, an observed image can be roughly decomposed into three parts (Figure 7U,S,N). Figure 7Y is from the rice grain image in the quiet area of the sun taken by the high-resolution imaging terminal of the one-meter infrared solar tower of Yunnan Observatory, Chinese Academy of Sciences.

$$Y = U + S + N. \tag{2}$$

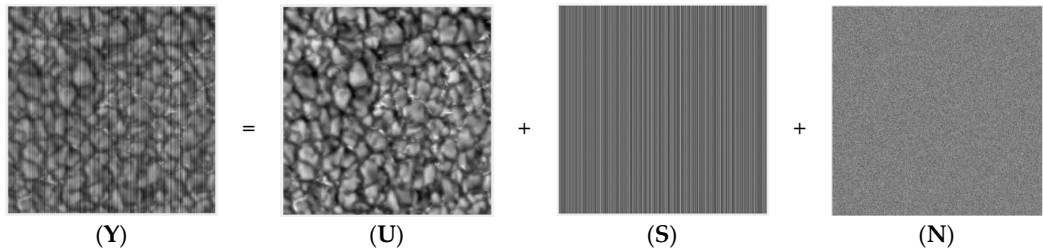

| (Y) | (U) | (S) | (N) |

**Figure 7.** Noise image composition. (**Y**) noise image; (**U**) clear image; (**S**) column fixed pattern noise (CFPN); (**N**) random noise.

Y is a noise image, U is a clear image to be estimated, N is the comprehensive image composed of PFPN, dark current, Poisson, and other random noises, and S is a CFPN. We need to estimate the values of S and U from Equation (2). Observations show that the equation is an underdetermined equation with more unknown quantities than the given data, but it can be converted into a well-defined equation with reasonable prior constraint information added. How to find constraint information, model, and solve the equation will be the key to this article. By analyzing the noise image, the following clues can be found:

1.  The specific directional structure of CFPN.

The vertical gradient histogram of noise image is very similar to that of clear image. From Figure 8 (histogram probability distribution of Figure 7Y,U), it can be found that the vertical gradient histogram distribution of the two images is very similar. The vertical gradient similarity between the two can be guaranteed by using the sparse constraint term of $\|\frac{\partial Y}{\partial y} - \frac{\partial U}{\partial y}\|_1$.

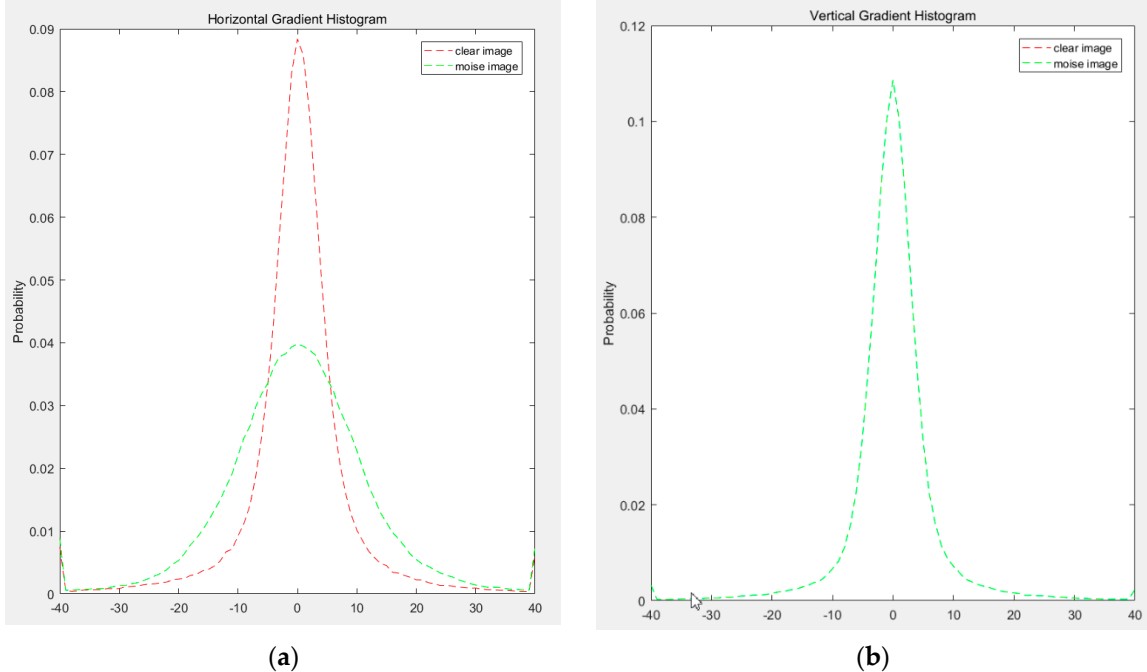

**Figure 8.** Gradient probability distribution. (**a**) Horizontal gradient probability distribution; (**b**) vertical gradient probability distribution (x coordinates: horizontally or vertically gradient; y coordinates: probability distribution).

2. The Striped noise has the characteristic of low rank.

Through the analysis of the CMOS output structure, it can be found that each column shares an output amplifier. Therefore, low rank can be used to describe the characteristics of CFPN.

3. Gaussian distribution of CFPN

$\Delta V_{A_{out}}$ has the characteristic of random Gaussian distribution [36], so we can add $\|S\|_2^2$ to express this random characteristic.

4. Structural similarity before and after noise removal

In order to maintain the structural similarity before and after noise removal, the constraint of 2 norm fidelity term $\|Y - U - S\|_2^2$ can be added.

5. Minimum variation of clear image

According to the correlation theory of total variation denoising, we know that the clear image has the characteristics of low variation value [37]. The regular term constraint of $\alpha_2\|\frac{\partial U}{\partial x}\|_1 + \alpha_3\|\frac{\partial U}{\partial y}\|_1$ can be added.

## 4. Proposed Model

Based on the fidelity items analyzed above and the relevant priori constraint information, the optimization equation with constraints can be obtained as follows:

$$\begin{aligned}
\text{E(U,S)} = \min_{U,S} \tfrac{\alpha_1}{2}\|Y - U - S\|_2^2 + \tau\|S\|_* \\
\text{s.t. } \|\tfrac{\partial U}{\partial x}\|_1 = 0 \\
\|\tfrac{\partial U}{\partial y}\|_1 = 0 \\
\|\tfrac{\partial Y}{\partial y} - \tfrac{\partial U}{\partial y}\|_1 = 0 \\
S\|_2^2 = 0
\end{aligned} \tag{3}$$

where $Y$ is the observed image, $U$ is the clear image and S is the stripe noise. $\|S\|_*$ is the kernel norm of S, which is used to constrain $S$ with low rank. Then, according to the Lagrange multiplier method, the constrained Equation (3) becomes an unconstrained Equation (4).

$$E(U,S) = \frac{\alpha_1}{2}\|Y - U - S\|_2^2 + \alpha_2\|\frac{\partial U}{\partial x}\|_1 + \alpha_3\|\frac{\partial U}{\partial y}\|_1 + \alpha_4\|\frac{\partial Y}{\partial y} - \frac{\partial U}{\partial y}\|_1$$
$$+ \frac{\gamma}{2}\|S\|_2^2 + \tau\|S\|_*$$
(4)

where $\alpha_1, \alpha_2, \alpha_3, \alpha_4, \gamma, \tau$ are Lagrange multipliers This is a multivariable convex optimization equation. At present, the commonly used methods for multivariable optimization are Bregman and ADMM. Through these optimization methods, the complex multivariable optimization process can be divided into more convenient sub optimization problems. The ADMM algorithm [38] is used in this paper. The specific optimization process is as follows: Three auxiliary variables H, J, K are introduced. Let $H = \frac{\partial U}{\partial x}$, $J = \frac{\partial U}{\partial y}$, $K = \frac{\partial Y}{\partial y} - \frac{\partial U}{\partial y}$, so Equation (4) equivalent to Equation (5)

$$E(U,S) = \frac{\alpha_1}{2}\|Y - U - S\|_2^2 + \alpha_2\|\frac{\partial U}{\partial x}\|_1 + \alpha_3\|\frac{\partial U}{\partial y}\|_1 + \alpha_4\|\frac{\partial Y}{\partial y} - \frac{\partial U}{\partial y}\|_1$$
$$+ \frac{\gamma}{2}\|S\|_2^2 + \tau\|S\|_*$$
$$\text{s.t. } H = \frac{\partial U}{\partial x}$$
$$J = \frac{\partial U}{\partial y}$$
$$K = \frac{\partial Y}{\partial y} - \frac{\partial U}{\partial y}$$
(5)

According to the Augmented Lagrange multiplier method, the constraint equation of Equation (5) is changed into the unconstrained Equation (6)

$$E(U,S,H,J,K) = \frac{\alpha_1}{2}\|Y - U - S\|_2^2 + \alpha_2\|H\|_1 + \alpha_3\|J\|_1 + \alpha_4\|K\|_1 + \frac{\gamma}{2}\|S\|_2^2 + \tau\|S\|_*$$
$$+ <R_2,\ H - \frac{\partial U}{\partial x}> + <R_3,\ J - \frac{\partial U}{\partial y}> + <R_4, K - \left(\frac{\partial Y}{\partial y} - \frac{\partial U}{\partial y}\right)>$$
$$+ \frac{\omega_2}{2}\|H - \frac{\partial U}{\partial x}\|_2^2 + \frac{\omega_3}{2}\|J - \frac{\partial U}{\partial y}\|_2^2 + \frac{\omega_4}{2}\|K - \left(\frac{\partial Y}{\partial y} - \frac{\partial U}{\partial y}\right)\|_2^2$$
(6)

where $<A, B>$ is defined as the inner product of two variables. $R_2$, $R_3$, $R_4$ are the regular coefficients of the regular terms $H - \frac{\partial U}{\partial x}$, $J - \frac{\partial U}{\partial y}$, $K - \left(\frac{\partial Y}{\partial y} - \frac{\partial U}{\partial y}\right)$, respectively.

Equation (6) can be changed into Equation (7) after the relevant terms are combined.

$$E(U,S,H,J,K) = \frac{\alpha_1}{2}\|Y - U - S\|_2^2 + \alpha_2\|H\|_1 + \alpha_3\|J\|_1 + \alpha_4\|K\|_1 + \frac{\gamma}{2}\|S\|_2^2 + \tau\|S\|_*$$
$$+ \frac{\omega_2}{2}\|H - \frac{\partial U}{\partial x} + \frac{R_2}{\omega_2}\|_2^2 + \frac{\omega_3}{2}\|J - \frac{\partial U}{\partial y} + \frac{R_3}{\omega_3}\|_2^2 + \frac{\omega_4}{2}\|K - \left(\frac{\partial Y}{\partial y} - \frac{\partial U}{\partial y}\right) + \frac{R_4}{\omega_4}\|_2^2$$
(7)

1. Sub-questions about U

$$\min_{U} E(U,S,H,J,K) = \frac{\alpha_1}{2}\|Y - U - S\|_2^2 + \frac{\omega_2}{2}\|H - \frac{\partial U}{\partial x} + \frac{R_2}{\omega_2}\|_2^2 + \frac{\omega_3}{2}\|J - \frac{\partial U}{\partial y} + \frac{R_3}{\omega_3}\|_2^2$$
$$+ \frac{\omega_4}{2}\|K - \left(\frac{\partial Y}{\partial y} - \frac{\partial U}{\partial y}\right) + \frac{R_4}{\omega_4}\|_2^2$$
(8)

2. Solve the extreme value about U

$$\frac{\partial E(U,S,H,J,K)}{\partial U} = 0$$
$$-\alpha_1(Y - U - S) - \omega_2\left(\frac{\partial H}{\partial x} - \frac{\partial^2 U}{\partial x^2} + \frac{1}{\omega_2}\frac{\partial R_2}{\partial x}\right) - \omega_3\left(\frac{\partial J}{\partial y} - \frac{\partial^2 U}{\partial y^2} + \frac{1}{\omega_3}\frac{\partial R_3}{\partial y}\right) + \omega_4\left(\frac{\partial K}{\partial y} - \left(\frac{\partial^2 Y}{\partial y^2} - \frac{\partial^2 U}{\partial y^2}\right) + \frac{1}{\omega_4}\frac{\partial R_4}{\partial y}\right) = 0$$

To make full use of the advantages of fast FFT calculation, we transform the formulas above into frequency domain to solve the equation. The specific calculation process is as follows: Simultaneous Fourier Transform on Both Sides of Equation

$$\mathcal{F}\left(\alpha_1 U + \omega_2 \frac{\partial^2 U}{\partial x^2} + \omega_3 \frac{\partial^2 U}{\partial y^2} + \omega_4 \frac{\partial^2 U}{\partial y^2}\right) =$$
$$\mathcal{F}\left(\alpha_1 Y - \alpha_1 S + \omega_2 \frac{\partial H}{\partial x} + \frac{\partial R_2}{\partial x} + \omega_3 \frac{\partial J}{\partial y} + \frac{\partial R_3}{\partial y} - \omega_4 \frac{\partial K}{\partial y} + \omega_4 \frac{\partial^2 Y}{\partial y^2} - \frac{\partial R_4}{\partial y}\right)$$
$$\mathcal{F}(BU) = \mathcal{F}(A)$$

where $A$ and $B$ are

$$A = \alpha_1 Y - \alpha_1 S + \omega_2 \frac{\partial H}{\partial x} + \frac{\partial R_2}{\partial x} + \omega_3 \frac{\partial J}{\partial y} + \frac{\partial R_3}{\partial y} - \omega_4 \frac{\partial K}{\partial y} + \omega_4 \frac{\partial^2 Y}{\partial y^2} - \frac{\partial R_4}{\partial y}$$

$$B = a_1 + \omega_2 \frac{\partial^2}{\partial x^2} + \omega_3 \frac{\partial^2}{\partial y^2} + \omega_4 \frac{\partial^2}{\partial y^2}$$

$$\mathcal{F}(U) = \frac{\mathcal{F}(A)}{\mathcal{F}(B)}$$

$$U = \mathcal{F}^{-1}\left(\frac{\mathcal{F}(A)}{\mathcal{F}(B)}\right) \tag{9}$$

where $\mathcal{F}(A)$ and $\mathcal{F}(B)$ are

$$\mathcal{F}(A) = \alpha_1 \mathcal{F}(Y - S) + \omega_2 \mathcal{F}\left(\frac{\partial}{\partial x}\right)\mathcal{F}(H) + \mathcal{F}\left(\frac{\partial}{\partial x}\right)\mathcal{F}(R_2) + \omega_3 \mathcal{F}\left(\frac{\partial}{\partial y}\right)\mathcal{F}(J) + \mathcal{F}\left(\frac{\partial}{\partial y}\right)\mathcal{F}(R_3)$$
$$-\omega_4 \mathcal{F}\left(\frac{\partial}{\partial y}\right)\mathcal{F}(K) + \omega_4 \mathcal{F}\left(\frac{\partial^2}{\partial y^2}\right)\mathcal{F}(Y) - \mathcal{F}\left(\frac{\partial}{\partial y}\right)\mathcal{F}(R_4)$$
$$\mathcal{F}(B) = \alpha_1 + \omega_2 \mathcal{F}\left(\frac{\partial^2}{\partial x^2}\right) + \omega_3 \mathcal{F}\left(\frac{\partial^2}{\partial y^2}\right) + \omega_4 \mathcal{F}\left(\frac{\partial^2}{\partial y^2}\right)$$

where $\mathcal{F}$ is the forward Fourier transform, $\mathcal{F}^{-1}$ is the inverse Fourier transform.

3.  Sub-questions about $S$

$$\min_S E(U, S, H, J, K) == \frac{\alpha_1}{2}\|Y - U - S\|_2^2 + \frac{\gamma}{2}\|S\|_2^2 + \tau\|S\|_* \tag{10}$$

The extremum solution process of the function with kernel norm can be summed up in two steps. First, the extremum of the non-kernel norm is solved, and then the dimension of S is reduced by singular value decomposition. The specific process is as follows:

- Step 1, solving the extreme value of non-kernel norm term in Formula (10)

$$\min_S E(U, S, H, J, K) == \frac{\alpha_1}{2}\|Y - U - S\|_2^2 + \frac{\gamma}{2}\|S\|_2^2 \tag{11}$$

Solve the extreme value about S

$$\frac{\partial E(U,S,H,J,K)}{\partial S} = -\alpha_1(Y - U - S) + \gamma S = 0$$
$$S = \frac{\alpha_1(Y-U)}{\alpha_1+\gamma}$$

- Step 2, reducing dimension of S by the soft threshold method

$$S = UShrink(D, \tau)V^T \tag{12}$$

where $Shrink(D, \tau) = diag\{[D(1:n) : zeros(n_{max} - n)]\}$, $U$ is the left singular matrix of $S$, $V$ is the right singular matrix of $S$, and $D$ is the diagonal matrix of $S$. Reduce the dimension of diagonal matrix $D$ by the following formula:

$$Shrink(D, \tau) = diag\{[D(1:n) : zeros(n_{max} - n)]\}$$

where $n_{max}$ is the total number of diagonal elements of the diagonal matrix $D$.

$$M = \sum_{i=1}^{i=n_{max}} D_{ii}$$

$$N = \sum_{i=1}^{i=n} D_{ii}$$

$$n = floor\left(\frac{N}{M} = \tau\right)$$

Among them, $\tau$ is used to control the degree of retention of principal components during SVD decomposition. This process is a process of rank reduction, in order to achieve the purpose of low rank.

4. Sub-questions about $H$

$$E(U, S, H, J, K) = \alpha_2 \|H\|_1 + \frac{\omega_2}{2} \left\| H - \frac{\partial U}{\partial x} + \frac{R_2}{\omega_2} \right\|_2^2 \tag{13}$$

Solve the extreme value about $H$.

$$\frac{\partial E(U, S, H, J, K)}{\partial H} = 0$$

$$H = \begin{cases} \frac{\partial U}{\partial x} - \frac{R_2}{\omega_2} - \frac{\alpha_2}{\omega_2}, & if \ \frac{\partial U}{\partial x} - \frac{R_2}{\omega_2} > \frac{\alpha_2}{\omega_2} \\ 0, & if \ \left| \frac{\partial U}{\partial x} - \frac{R_2}{\omega_2} \right| \le \frac{\alpha_2}{\omega_2} \\ \frac{\partial U}{\partial x} - \frac{R_2}{\omega_2} + \frac{\alpha_2}{\omega_2}, & if \ \frac{\partial U}{\partial x} - \frac{R_2}{\omega_2} < -\frac{\alpha_2}{\omega_2} \end{cases} \tag{14}$$

5. Sub-questions about $J$

$$(U, S, H, J, K) = \alpha_3 \|J\|_1 + \frac{\omega_3}{2} \left\| J - \frac{\partial U}{\partial y} + \frac{R_3}{\omega_3} \right\|_2^2 \tag{15}$$

Solve the optimal solution of $J$.

$$\frac{\partial E(U, S, H, J, K)}{\partial J} = 0$$

$$J = \begin{cases} \frac{\partial U}{\partial y} - \frac{R_3}{\omega_3} - \frac{\alpha_3}{\omega_3}, & if \ \frac{\partial U}{\partial y} - \frac{R_3}{\omega_3} > \frac{\alpha_3}{\omega_3} \\ 0, & if \ \left| \frac{\partial U}{\partial y} - \frac{R_3}{\omega_3} \right| \le \frac{\alpha_3}{\omega_3} \\ \frac{\partial U}{\partial y} - \frac{R_3}{\omega_3} + \frac{\alpha_3}{\omega_3}, & if \ \frac{\partial U}{\partial y} - \frac{R_3}{\omega_3} < -\frac{\alpha_3}{\omega_3} \end{cases} \tag{16}$$

6. Sub-questions about $K$

$$E(U, S, H, J, K) = \alpha_4 \|K\|_1 + \frac{\omega_4}{2} \left\| K - \left( \frac{\partial Y}{\partial y} - \frac{\partial U}{\partial y} \right) + \frac{R_4}{\omega_4} \right\|_2^2 \tag{17}$$

Solve the optimal solution of *K*.

$$\frac{\partial E(U, S, H, J, K)}{\partial K} = 0$$

$$K = \begin{cases} \left(\frac{\partial Y}{\partial y} - \frac{\partial U}{\partial y}\right) - \frac{R_4}{\omega_4} - \frac{\alpha_4}{\omega_4}, & if \left(\frac{\partial Y}{\partial y} - \frac{\partial U}{\partial y}\right) - \frac{R_4}{\omega_4} > \frac{\alpha_4}{\omega_4} \\ 0, & if \left|\left(\frac{\partial Y}{\partial y} - \frac{\partial U}{\partial y}\right)\right| - \frac{R_4}{\omega_4} \leq \frac{\alpha_4}{\omega_4} \\ \left(\frac{\partial Y}{\partial y} - \frac{\partial U}{\partial y}\right) - \frac{R_4}{\omega_4} + \frac{\alpha_4}{\omega_4}, & if \left(\frac{\partial Y}{\partial y} - \frac{\partial U}{\partial y}\right) - \frac{R_4}{\omega_4} < -\frac{\alpha_4}{\omega_4} \end{cases} \tag{18}$$

7. Updating the language multiplier $R_2 \, R_3 \, R_4$ by the dual gradient rise method

$$R_2 = R_2 + \omega_2\left(H - \frac{\partial U}{\partial x}\right) \tag{19}$$

$$R_3 = R_3 + \omega_3\left(J - \frac{\partial U}{\partial y}\right) \tag{20}$$

$$R_4 = R_4 + \omega_4\left(K - \left(\frac{\partial Y}{\partial y} - \frac{\partial U}{\partial y}\right)\right) \tag{21}$$

where $\omega_2, \omega_3, \omega_4$ is the iterative step length in the process of gradient rise.

In order to facilitate computer computation, we need to discretize continuous operators. The discretization of partial differential is defined as follows: $\frac{\partial A}{\partial x}$ discrete operation is $A_{ij+1} - A_{ij}$, $\frac{\partial A}{\partial y}$ discrete operation is $A_{i+1j} - A_{ij}$, $\frac{\partial^2 A}{\partial x^2}$ discrete operation is $A_{ij+1} - A_{ij-1} - 2A_{ij}$, $\frac{\partial^2 A}{\partial y^2}$ discrete operation is $A_{i+1j} - A_{i-1j} - 2A_{ij}$. The complete calculation process is shown in Algorithm 1. The matlab code used in this article can be obtained from the download link in Supplementary Materials.

---

**Algorithm 1** Low-Rank sparase variationnal destripe (LRSUTV)

---

1. Get image Y with FPN
2. The initial matrix **U = 0, S = 0, $R_2$ = 0, $R_3$ = 0, $R_4$ = 0, H = 0, J = 0, K = 0**
3. Initial optimization factor $\alpha_1$, $\alpha_2$, $\omega_1$, $\omega_2$, $\omega_3$, $\omega_4$, $\tau$, **N**
4. For *n* = 1:N do
5. Calculating the optimal solution of U via Fourier Transformation by (9)
6. Calculating low rank S by singular value decomposition (SVD) by (12) (13)
7. calculating H J K through soft thresholds by (14), (16), (18)
8. **Update** $R_2 \, R_3 \, R_4$, by method of dual gradient rise by (19), (20), (21)
9. End for
10. Separate clear image U and stripe S

---

## 5. Experimental Results and Discussions

### 5.1. Experimental Environment

Before the experiment, to facilitate the display of the image, we encoded the original image into the gray scale of [0.255], and set the CFPN with standard deviation intensity in the [0.20] range. For scientific CIS, the general photo response nonuniformity (PRNU) is about 0.5%, and for consumer CMOS, it is about 2%. We set the noise intensity in the range of [0.20], so that the noise level of these two kinds of CIS can be completely expressed. To illustrate the effectiveness of the proposed algorithm (LRSUTV), we tested it from both the simulation and real data. In this paper, six scene-based FPN correction algorithms are selected for comparison experiments. They are wavelet [17,39,40], anisotropic total variation (UTV) [41–43], ASSTV [29], variational stationary noise remover (SNR), SILR [28], $\ell_0$ sparse method ($\ell_0$ sparse) [34], and the recommended method in this work.

To comprehensively and objectively reflect the correction effect of FPN, we use several common quality evaluation methods to evaluate the noise removal effect of each algorithm, namely Mean Cross-track (*x*-axis stands for the column number of the image, and the *y*-axis represents the mean value of each column), PSNR and SSIM. The various Lagrange multipliers used in our algorithm are hard-tuned. In order to objectively compare the effects of the various methods, I have adjusted the parameters of all methods to the best of my ability.

### 5.2. Simulation Experiment

In the simulation experiment, we generated the noise image of CFPN with different intensities. In order to restore the fact that CFPN has the characteristics of zero mean Gaussian distribution, we set the mean value $\mu = 0$, $\sigma = [0:20]$ of CFPN to test the denoising performance of various algorithms. In terms of the sCMOS cameras we currently use, almost all the CFPN we encountered presented aperiodic random stripes, and the PRNU is generally around 0.5%. However, some CMOS cameras used in some fields have periodic stripes. Therefore, in the comparison test of algorithms, we also simulated the existence of periodic noise. Next, we compared and evaluated the effectiveness of our method from two aspects: aperiodic and periodic noise. In terms of picture selection, we chose two pictures, one of which was a picture of a solar active area with a rich structure, and the other being a picture of a relatively single structure of the solar sphere. These two types of pictures are the types of images often encountered in solar observation.

a.    Aperiodic stripe noise

In the simulation experiment of aperiodic stripe, we generated a set of random column noise with mean value $\mu = 0$ and standard deviation $\sigma = [0:20]$, and the position of noise is random.

From the test results of Figure 9, the WAFT method and ASSTV method corresponding to Figure 9d,g achieved the worst denoising effect, and obvious stripes remained after denoising. Figure 9e has better results than Figure 9d,g, almost removing all stripes, except for some areas with wider stripes. In these areas, UTV showed incomplete stripe removal, with a certain degree of residue. From a visual point of view, several methods corresponding to Figure 9f,h–j achieved the best results. They completely removed all the stripes. Visually, it is difficult for me to distinguish the differences.

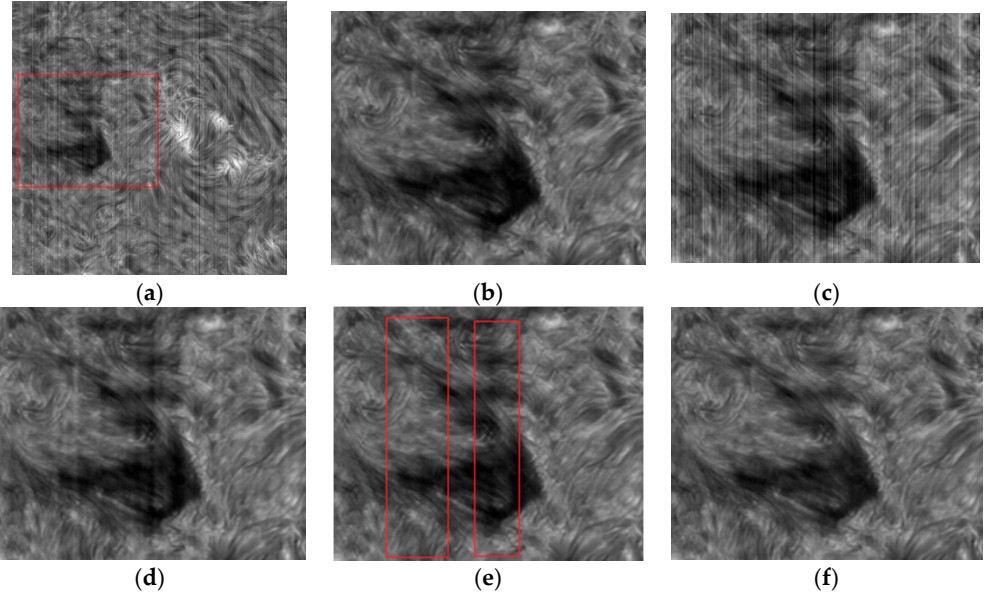

(a)    (b)    (c)

(d)    (e)    (f)

**Figure 9.** *Cont.*

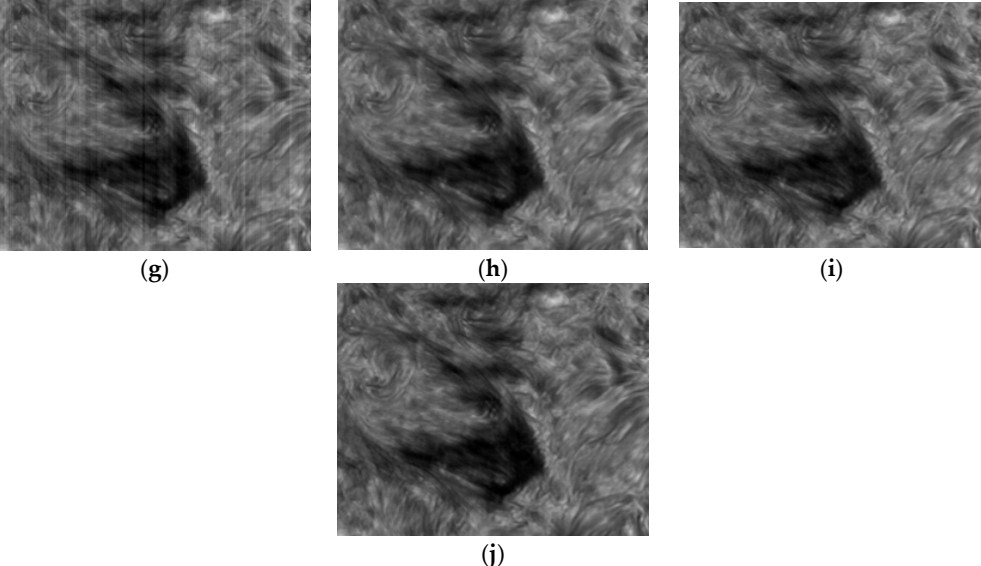

**Figure 9.** The results of various methods for removing stripes in the solar active area. (**a**) Global image with noise; (**b**) Original image of framed area; (**c**) Noise image of framed area; (**d**) WAFT; (**e**) Anisotropic total variation (UTV); (**f**) VSNR; (**g**) ASSTV; (**h**) SILR; (**i**) $\ell_0$ sparse; (**j**) LRSUTV.

Subjective Qualitative Evaluation

Next, to further distinguish the differences between VSNR, SILR, $\ell_0$ sparse, and LRSUTV's denoising results, we first made a qualitative comparison of the various methods using the difference image formed by the difference between the original image and the denoising result, and the mean cross-track curve of the denoising result. We then used the PSNR and SSIM values of various denoising results for a quantitative comparison.

The difference images shown in Figure 10 clearly show the stripe extraction ability of various methods, and whether the various methods damage the original image structure during the stripe extraction process.

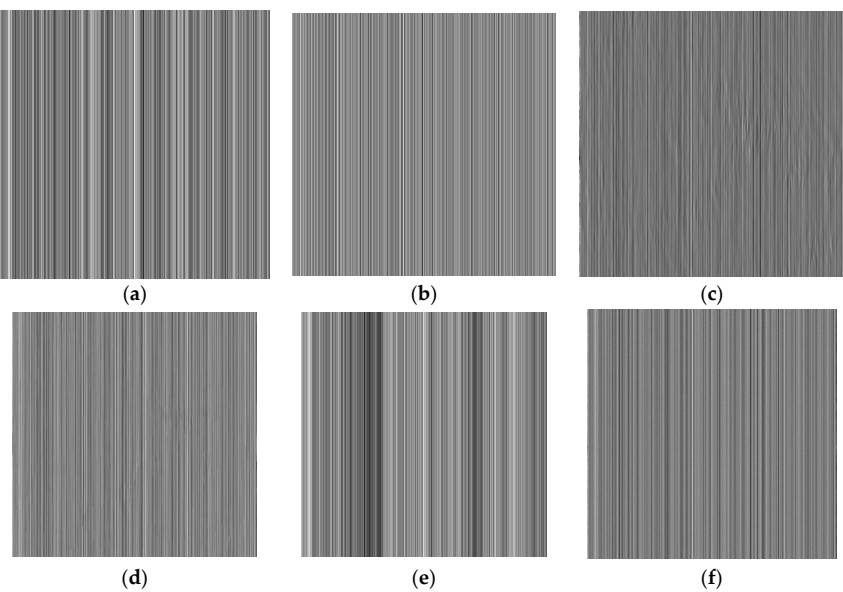

**Figure 10.** *Cont.*

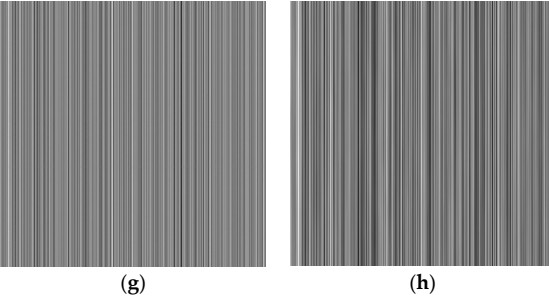

(**g**)                    (**h**)

**Figure 10.** Stripes extracted by various methods. (**a**) Added CFPN; (**b**) WAFT; (**c**) UTV; (**d**) ASSTV; (**e**) VSNR; (**f**) SILR; (**g**) $\ell_0$ sparse; (**h**) LRSUTV.

It can be found from Figure 10b that the stripe extracted by WAFT has a large error compared to Figure 10a, which indicates that the image after WAFT denoising still has a lot of residual noise stripe. This situation can also be observed in Figure 9d. The stripes in Figure 10c,d are similar to those in Figure 10a, but there is a certain degree of residual image structure information in the stripes. This shows that the denoising results of UTV and ASSTV in these residual areas have a certain degree of damage to the original structure. Figure 10e has a high similarity with Figure 10a, but some areas are bright, which will cause the brightness of the VSNR denoised image in this area to be darker than the original image. Figure 10f,g has a very high similarity to Figure 10a, but there is a shift in overall intensity. The shift in intensity causes the brightness and darkness of the denoised image to be different from the original image. Then, the stripe information extracted by our proposed LRSUTV method is the closest to Figure 10a both in structure and intensity.

The mean cross-track curve is also a commonly used image quality evaluation method, through which the overall trend of the image can be clearly observed. Next, we conducted a qualitative evaluation of various results through the mean cross-track curve. It can be seen from Figure 11 that the curve of Figure 11b,d is significantly different from that of Figure 11a, and there is a significant curve fluctuation caused by incomplete stripe removal. This conclusion is fully consistent with the subjective feeling of Figure 9. The curve of Figure 11e has obvious deviations in some areas compared to the original curve. After comparing Figure 10a, it was found that the CFPN in these areas has a certain width. It can be inferred that the VSNR denoising method will produce a certain error when processing wide stripes. The overall situation of the Figure 11c curve is better than that of Figure 11b,d,e, but there is excessive smoothness, which means that the details of the image are lost. Figure 11a,g are very similar in appearance.

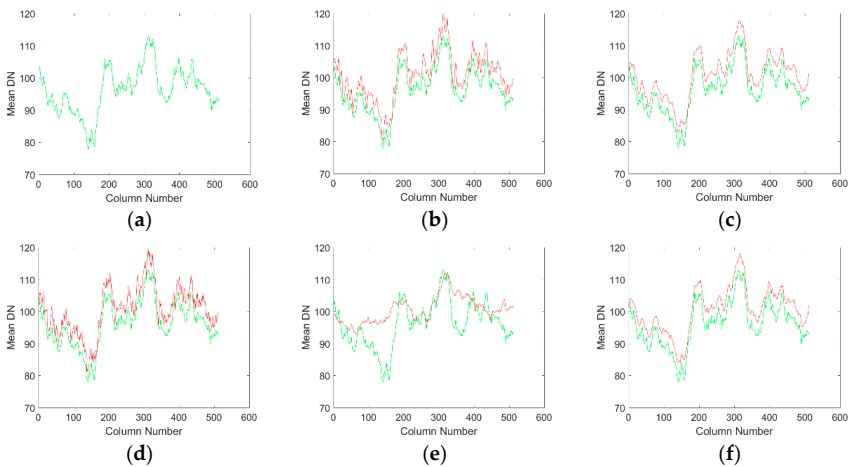

**Figure 11.** *Cont.*

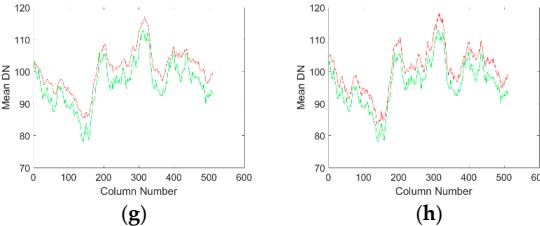

(g)                    (h)

**Figure 11.** Mean cross-track curves of various denoising results, where the green curve in each result is the mean cross-track curve of the original image, and the red curve is the mean cross-track curve of various denoising results. (**a**) Original image; (**b**) WAFT; (**c**) UTV; (**d**) ASSTV; (**e**) VSNR; (**f**) SILR; (**g**) $\ell_0$ sparse; (**h**) LRSUTV.

When you observe carefully, you will find that the curve of Figure 11g has an overall upward shift, which means that the denoised image has an overall shift in brightness compared to the original image. The curves of Figure 11f,h are very similar to Figure 11a in terms of strength and shape, but Figure 11f has a certain degree of excessive smoothing, while Figure 11h retains more details.

*5.3. Periodic Stripe Noise*

In a similar way, we then observed a picture of the solar sphere with a relatively simple structure. In order to analyze the ability of various methods to remove periodic noise, we added a strip noise with period T = 16, mean μ = 0 and σ = 15 on this picture. As can be seen from Figure 12, except for Figure 12d,g, the rest of the pictures have a very good stripe removal effect, and have a very similar structure to Figure 12b. Therefore, we also needed to use the stripe extraction image and mean cross-track curve to further judge the advantages and disadvantages of various methods.

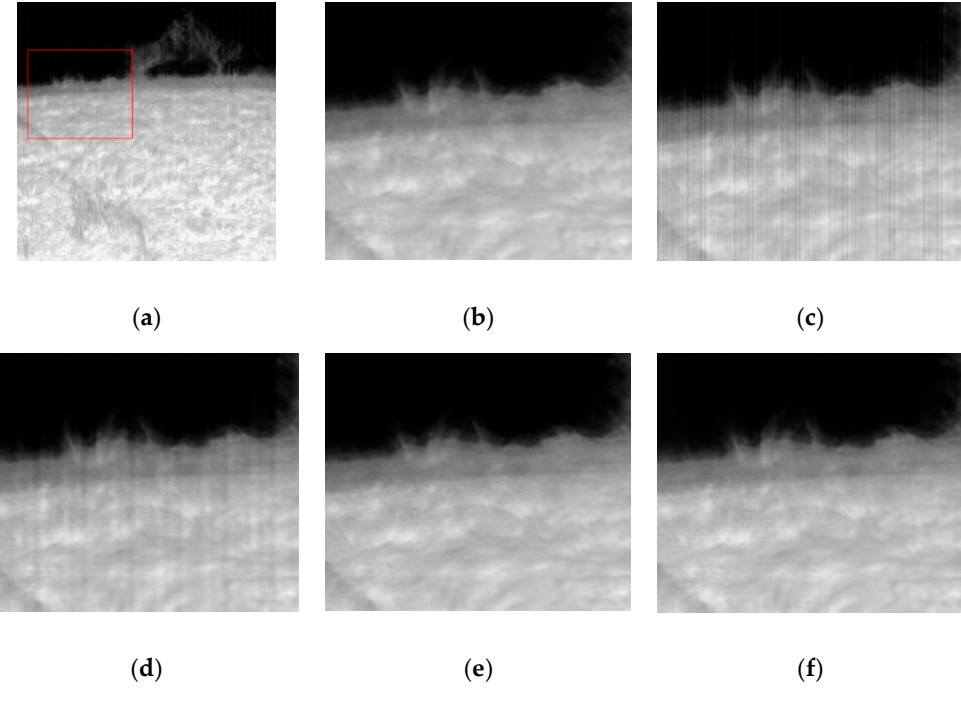

(a)                    (b)                    (c)

(d)                    (e)                    (f)

**Figure 12.** *Cont.*

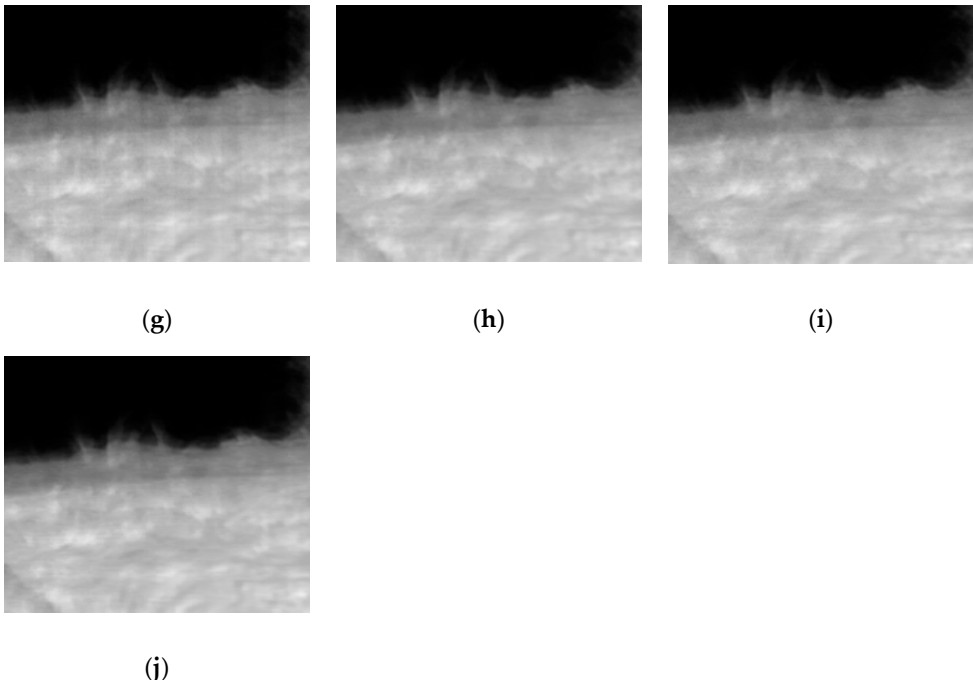

(g)          (h)          (i)

(j)

**Figure 12.** Various methods of stripe removal result on the solar sphere image. (**a**) Global image with noise; (**b**) the original image of the comparison area; (**c**) the noise image of the comparison area; (**d**) WAFT; (**e**) UTV; (**f**) VSNR; (**g**) ASSTV; (**h**) SILR; (**i**) $\ell_0$ sparse; (**j**) LRSUTV.

Subjective Qualitative Evaluation

From the results of Figure 13 stripe extraction, WAFT has the worst effect, and the extracted stripes are not similar to Figure 13a. Figure 13c,d,g are the same as Figure 13a. There is a certain degree of similarity, but the extracted results still carry weak original image structure information. This means that the UTV, ASSTV, $\ell_0$ sparse methods still have streaks left after denoising, but the residual amplitude is not strong enough to make it difficult for the human eye to distinguish. From the streaks in Figure 13e, you can see the periodic trend. However, there is a case where the intensity value is obviously large in some areas, which indicates that the VSNR has a small estimate of the fringes in this area. Figure 13f,h has the highest similarity to Figure 13a, but it can also be clearly seen that the SILR denoising result has the fact that the overall intensity is relatively small. Overall, the results of LRSUTV are closest to the original CFPN, and the effect of stripe extraction is the best.

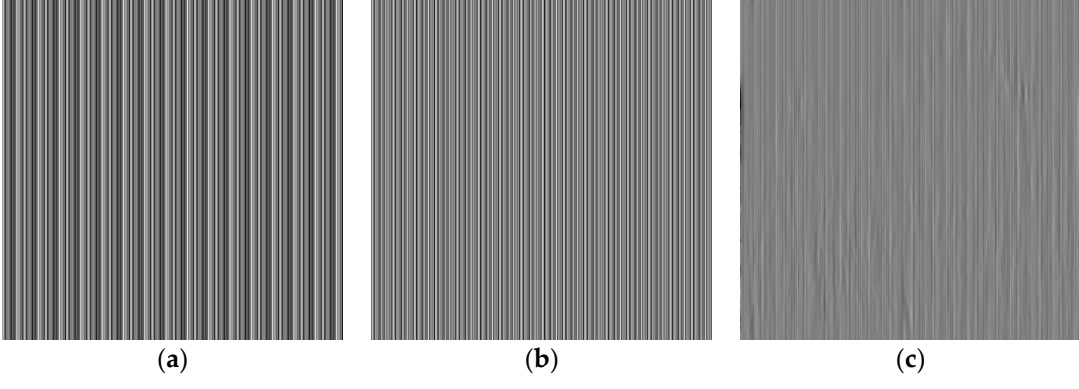

(a)          (b)          (c)

**Figure 13.** *Cont.*

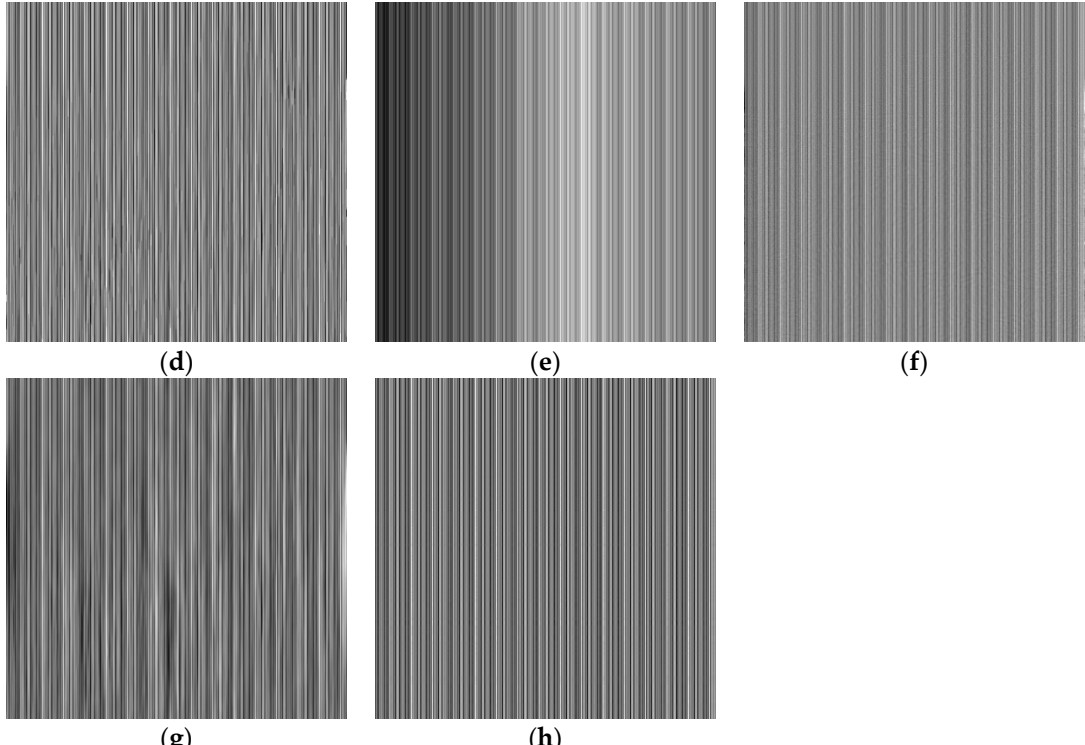

**Figure 13.** Stripes extracted by various methods. (**a**) Added CFPN; (**b**) WAFT; (**c**) UTV; (**d**) ASSTV; (**e**) VSNR; (**f**) SILR; (**g**) $\ell_0$ sparse; (**h**) LRSUTV.

The same conclusion can be found from the analysis of the mean cross-track curve. The curve of Figure 14f–h is most similar to that of Figure 14a, but it can be found that Figure 14f,g has excessive smoothing, which will lead to the loss of detailed information. The current image structure is very simple, so it is difficult to visually detect the difference in their denoising results.

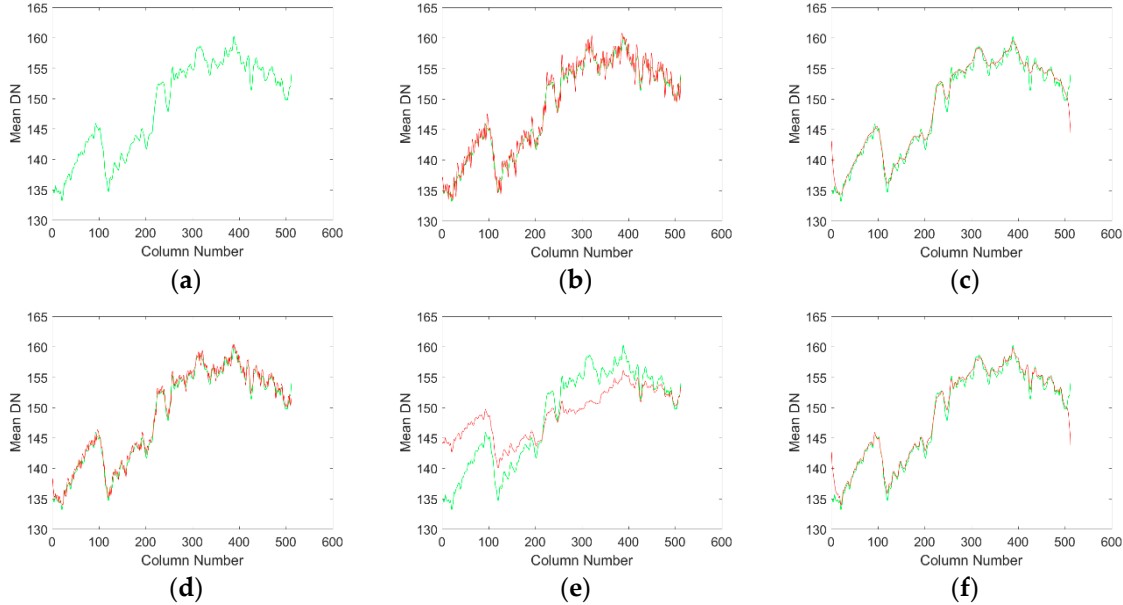

**Figure 14.** *Cont.*

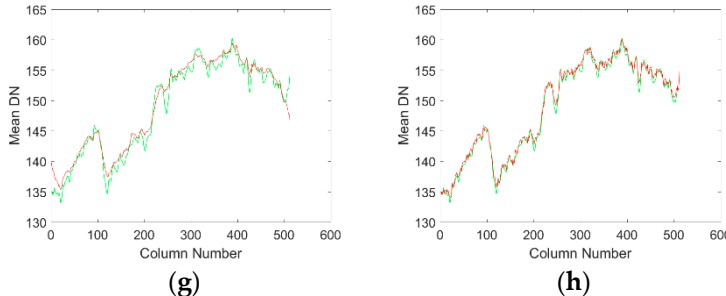

(g)                                          (h)

**Figure 14.** Mean cross-track curves of various denoising results, where the green curve in each result is the mean cross-track curve of the original image, and the red curve is the mean cross-track curve of various denoising results. (**a**) Original image; (**b**) WAFT; (**c**) UTV; (**d**) ASSTV; (**e**) VSNR; (**f**) SILR; (**g**) $\ell_0$ sparse; (**h**) LRSUTV.

Because the stripe noise is periodic, we can also compare the power spectrum curves of various denoising results and observe the suppression of noise pulse by various methods. The curve shown in Figure 15 is the power spectrum curve of Figure 12a. For better vision, we normalized the frequency of the *x*-axis and logarithmically calculated the power spectrum amplitude of the *y*-axis. Due to the periodicity of the noise, the power spectrum curve of Figure 12a shows an obvious pulse signal at some frequencies. After denoising, LRSUTV removes all obvious pulse signals, retains details at maximum range, and maintains the same spectral intensity as the original image. However, WAFT, UTV, ASSTV have distinct large pulse residues, which means striped residues. SILR and $\ell_0$ sparse have significant intensity differences compared with Figure 15a, which means that the overall brightness of the image is different from the original image. The power spectrum on the left side of Figure 15e is significantly different from that of the original, which means that the image after VSNR denoising will have noise residue in the low frequency area.

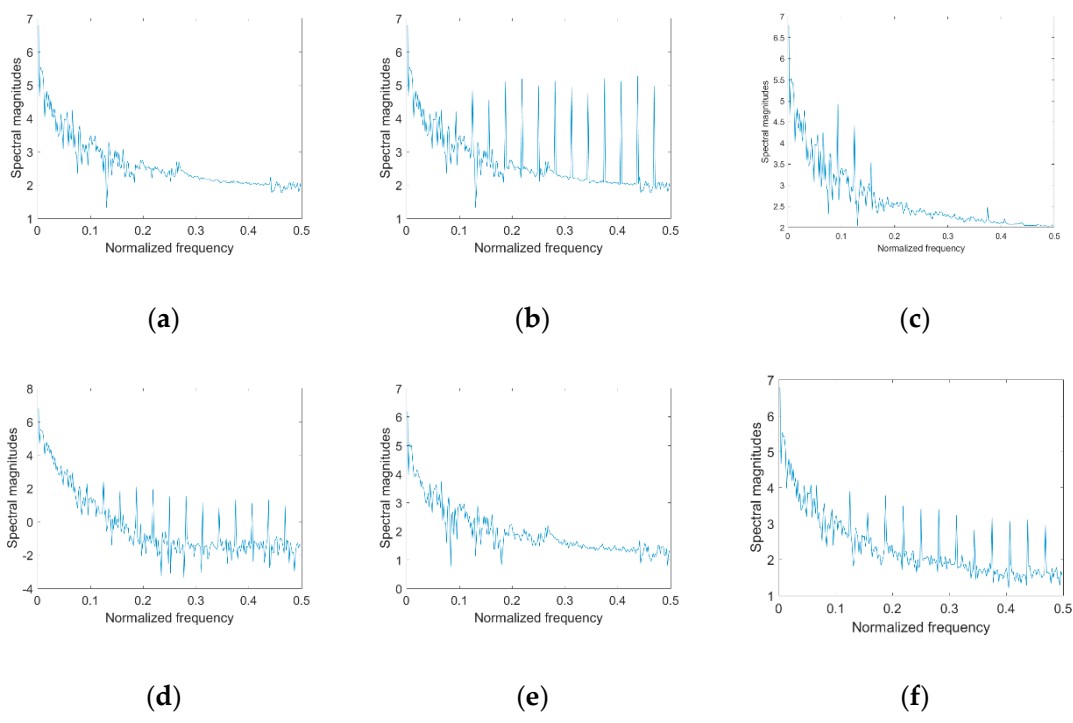

**Figure 15.** *Cont.*

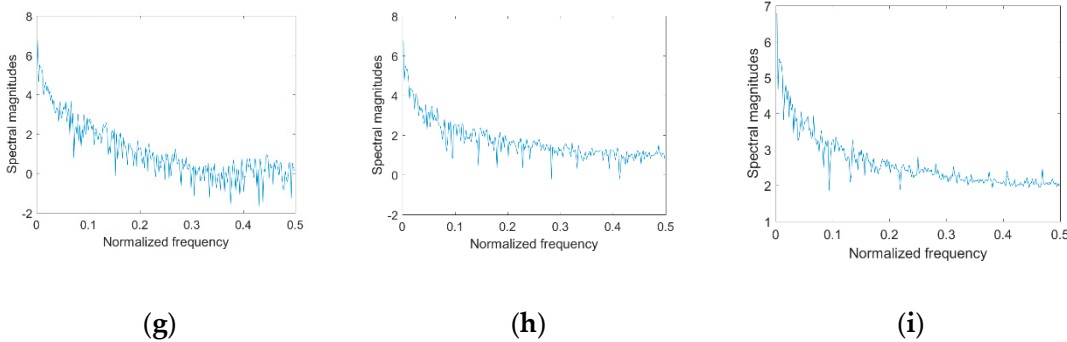

(g)                (h)               (i)

**Figure 15.** The power spectrum of the image in the solar active region. (**a**) Original image; (**b**) noise image; (**c**) WAFT; (**d**) UTV; (**e**)VSNR; (**f**) ASSTV; (**g**) SILR; (**h**) $\ell_0$ sparse; (**i**) LRSUTV.

### 5.4. Quantitative Objective Evaluation

The above analysis is a relatively subjective one. Different people may have different conclusions. Next, we quantitatively give the performance of various methods under different noise intensity and different images in a more objective way. Since we have the data from the original image, we will evaluate the result of the noise reduction using a full-reference approach. The evaluation indexes are: PSNR, SSIM. In Tables 1–4, is the standard deviation of the stripe noise. Pictures of the solar active region add aperiodic noise, and photosphere pictures add periodic noise with a period of 16.

**Table 1.** PSNR of denoising results in solar active region by various methods for solar active region.

| Images | Method | $\sigma = 4$ | $\sigma = 8$ | $\sigma = 12$ | $\sigma = 16$ | $\sigma = 20$ |
|---|---|---|---|---|---|---|
| | WAFT | 31.59958 | 31.34382 | 30.87000 | 30.50006 | 28.54466 |
| | UTV | 31.31853 | 31.16326 | 30.87973 | 30.82699 | 29.55368 |
| Solar Active | ASSTV | 31.61487 | 31.32085 | 30.76171 | 30.16115 | 27.63940 |
| Region | VSNR | 29.55233 | 29.45862 | 29.28992 | 29.38995 | 29.41818 |
| | SILR | 32.15904 | 32.06835 | 31.75536 | 31.81308 | 30.89457 |
| | L0 | 31.48390 | 31.21754 | 30.97054 | 31.10174 | 30.95841 |
| | LRSUTV | 32.39316 | 32.19943 | 31.86411 | 31.93278 | 31.05222 |

**Table 2.** PSNR of denoising results of various methods for solar granular.

| Images | Method | $\sigma = 4$ | $\sigma = 8$ | $\sigma = 12$ | $\sigma = 16$ | $\sigma = 20$ |
|---|---|---|---|---|---|---|
| | WAFT | 33.95483 | 33.70573 | 33.00279 | 31.98996 | 29.77545 |
| | UTV | 33.83339 | 33.79097 | 33.56776 | 33.27398 | 31.70765 |
| Solar | ASSTV | 34.03604 | 33.77074 | 33.04330 | 31.79459 | 29.08905 |
| photospheric | VSNR | 32.69314 | 32.72424 | 32.73020 | 32.76216 | 32.39804 |
| layer | SILR | 35.43197 | 35.46269 | 35.19106 | 34.89504 | 33.39456 |
| | L0 | 33.79005 | 33.69596 | 33.57622 | 33.38637 | 32.86435 |
| | LRSUTV | 36.80814 | 35.74191 | 35.52385 | 35.42470 | 33.80298 |

**Table 3.** SSIM of denoising results in solar active region by various methods.

| Images | Method | $\sigma = 4$ | $\sigma = 8$ | $\sigma = 12$ | $\sigma = 16$ | $\sigma = 20$ |
|---|---|---|---|---|---|---|
| | WAFT | 0.979698 | 0.976763 | 0.971173 | 0.959207 | 0.911204 |
| | UTV | 0.973915 | 0.973032 | 0.971825 | 0.968449 | 0.946812 |
| Solar active | ASSTV | 0.97961 | 0.976334 | 0.969964 | 0.954213 | 0.889726 |
| region | VSNR | 0.972853 | 0.972645 | 0.972404 | 0.972431 | 0.972587 |
| | SILR | 0.9813 | 0.980956 | 0.980165 | 0.97885 | 0.970024 |
| | L0 | 0.977923 | 0.977109 | 0.976615 | 0.976124 | 0.975494 |
| | LRSUTV | **0.98914** | **0.981068** | **0.980475** | **0.97893** | **0.976364** |

**Table 4.** SSIM of denoising results of various methods for solar granular.

| Images | Method | σ = 4 | σ = 8 | σ = 12 | σ = 16 | σ = 20 |
|---|---|---|---|---|---|---|
| Solar photospheric layer | WAFT | 0.942124 | 0.927306 | 0.897961 | 0.852761 | 0.768504 |
| | UTV | 0.941753 | 0.935944 | 0.927616 | 0.907744 | 0.853808 |
| | ASSTV | 0.942987 | 0.929386 | 0.902473 | 0.85379 | 0.74729 |
| | VSNR | 0.877748 | 0.878157 | 0.880637 | 0.87524 | 0.878135 |
| | SILR | 0.9517 | 0.945622 | 0.9384 | 0.922522 | 0.887166 |
| | L0 | 0.938884 | 0.927177 | 0.927061 | 0.915114 | 0.909504 |
| | LRSUTV | 0.957022 | 0.950072 | 0.940674 | 0.928795 | 0.895663 |

From Tables 1–4 above, we can see that our proposed LRSUTV method is superior to all the methods on the PSNR index. On the SSIM index, our method is superior to all other methods, except when the variance is 20. This is mainly due to the adaptive adjustment strategy with some parameters in our algorithm and the reasonable model structure. All the methods involved in the comparison adjust their parameters to the best when σ = 12. Then, they are applied to different noise levels for testing. Experiments show that our method can adapt to different noise levels, and has good robustness.

*5.5. Actual Image Testing*

Figure 16 is a sunspot image in the TiO band observed by the 1-m infrared solar tower of Yunnan Observatory. There is obvious CFPN noise on the surface of the observed image, which greatly reduces the quality of the image.

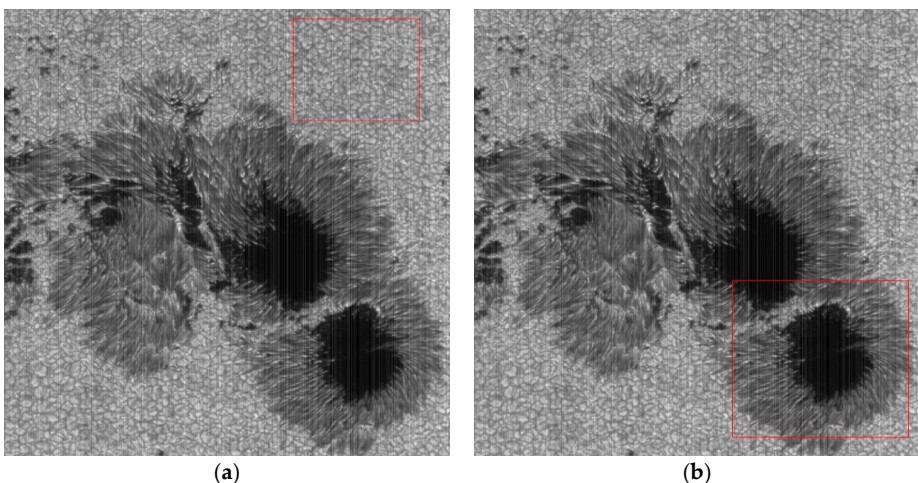

**Figure 16.** Photo of sunspots in the solar TiO band. (**a**) Emphasis area is granular areas; (**b**) emphasis area is sunspot.

From the observation of Figures 17 and 18, It is seen that both for the images of the solar granular and complex sunspots, there still exits the residual noise after the reduction of WAFT, UTV, and ASSTV. The result still has obvious stripe noise. Looking at Figure 18e, we can see that there is a distinct variable at the center of the sunspot, which is due to the incorrect estimation of the central zone stripes by VSNR. The observation of Figure 18f–h shows that all three can remove noise stripes thoroughly. Although all three methods remove stripes effectively, the SILR and $\ell_0$ sparse methods do not suppress random noise effectively. Significant random noise still exists in the result of denoising. This is mainly due to their simple assumption that noise images are the result of the overlay of clear images and stripe noise in the stage of model building, that is, $Y = U + S$, where Y is the noise image, U is the clear image, and S is the stripe noise image. In fact, $Y = U + S + N$, where N is the random noise caused

by the combination of reset noise, shot noise, thermal noise, Poisson noise, and so on, in the camera. Therefore, the LRSUTV denoising method is a more comprehensive one.

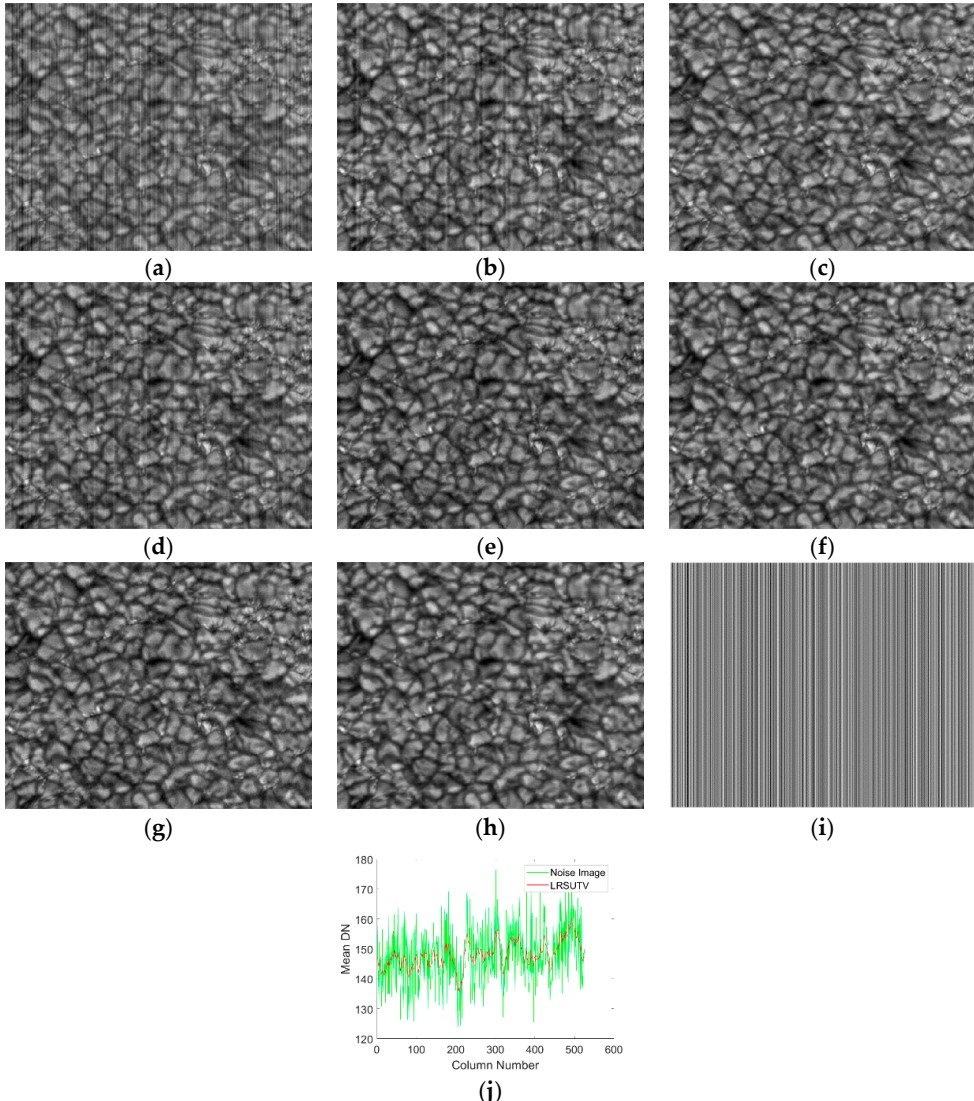

**Figure 17.** Denoising effect of various methods in the area of solar granular. (**a**) Noise Image; (**b**) WAFT; (**c**) UTV; (**d**) ASSTV; (**e**) VSNR; (**f**) SILR; (**g**) $\ell_0$ sparse; (**h**) LRSUTV; (**i**) CFPN noise estimated by LRSUTV method; (**j**) mean cross track of images (**a**,**h**).

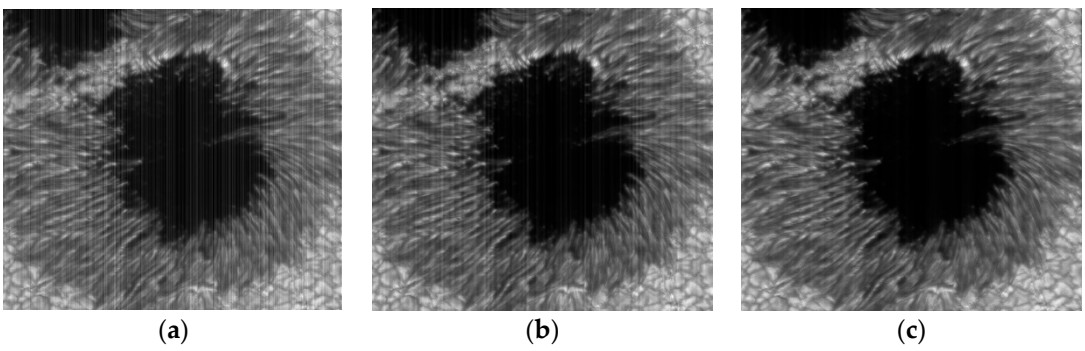

**Figure 18.** *Cont.*

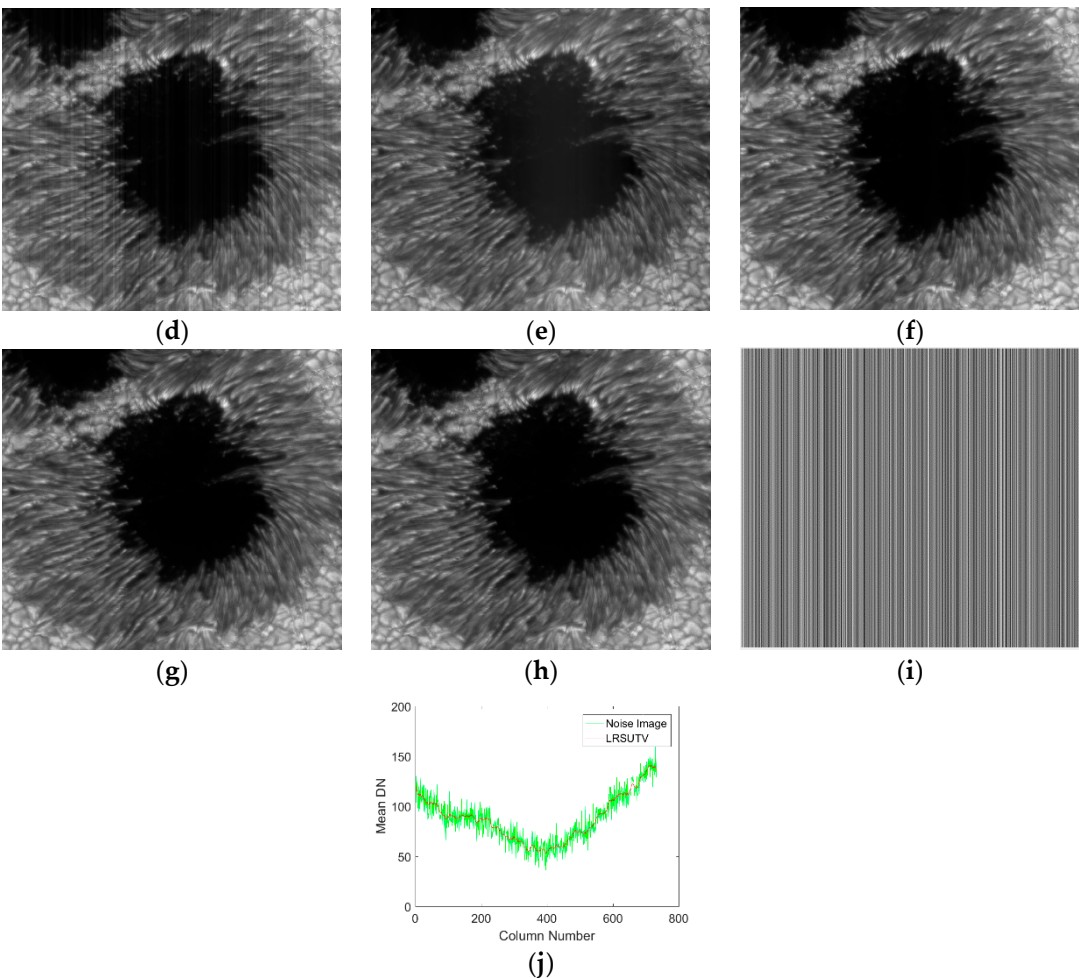

**Figure 18.** Denoising effect of various methods in the area of sunspot. (**a**) Noise Image; (**b**) WAFT; (**c**) UTV; (**d**) ASSTV; (**e**) VSNR; (**f**) SILR; (**g**) $\ell_0$ sparse; (**h**) LRSUTV; (**i**) CFPN noise estimated by LRSUTV method; (**j**) mean cross track of images (**a**,**h**).

## 5.6. Discussion

### 5.6.1. Parameter Selection

In Model (7), adjustment of parameters $\alpha_1, \alpha_2, \alpha_3, \alpha_4, \gamma, \tau$ are involved. The basic principle of parameter adjustment is that as the variance of stripe noise increases, the value of $\gamma$ needs to be increased accordingly. The best PSNR value is obtained by adjusting $\gamma$ between [0.65–0.85], when $\sigma$ changes between [5:20] ranges, according to experience. The adjustment for $\tau$ is generally between [0.5–0.75], based on experience. The adjustment of $\alpha_2$ is critical, because its values vary widely with the intensity of images and stripes. To simplify the adjustment of $\alpha_2$, we used an adaptive strategy. The basic steps are as follows:

First, the Fourier transform of the noise image Y in each column is calculated, as shown in Formula (22).

$$F_{:i} = \mathcal{F}(Y_{:i}) \tag{22}$$

where F represents the forward Fourier transform operator, Y represents the input noise image, and i represents the index of the column.

Second, update the regular parameter $\alpha_2$.

$$\alpha_2 = \frac{\|\frac{\partial F_{1:}}{\partial x}\|_1}{10^5} \cdot \alpha_1 \tag{23}$$

where $\frac{\partial F_{1:}}{\partial x}$ represents the horizontal differentiation of the DC component in F. The algorithm automatically calculates the relationship between $\alpha_2$ and the fidelity term regularization factor $\alpha_1$, based on the intensity of the current stripe noise. The $\alpha_1$ coefficient is set to a fixed value of $\alpha_1 = 20$ in the LRSUTV algorithm. The algorithm can automatically enhance the $\alpha_2$ parameter if the intensity of the stripe noise is too high, which will cause the stripe suppression process to dominate in the iteration process. As the intensity of the stripe decreases, the $\|\frac{\partial F_{1:}}{\partial x}\|_1$ value will become smaller and, thus, the fidelity process will dominate. As a rule of thumb, the values of $\alpha_3$ and $\alpha_4$ are set at [15:25] and [1.5:2.5], respectively. At this time, the denoised image has the optimal PSNR value. To simplify the parameter adjustment process, we generally set $\omega_2 = 0.1$, $\omega_3 = 0.1$ and $\omega_4 = 0.1$.

The parameters in Table 5 are the optimal parameters for various methods for Figure 9a. Figure 9a contains stripe noise with intensity_$\sigma = 10$ and random noise with intensity_$\sigma = 5$. By setting the above parameters, each method obtains the best PSNR for Figure 9a. The above optimization parameters are used for different pictures and different noise intensities used in the experiment.

**Table 5.** Parameter settings of various methods being compared.

| Method | Key Parameter |
|--------|---------------|
| WAFT | numlev $= 2$, wavtyp $='$ db7$'$, k $= 2.8$ |
| UTV | $\alpha = 500$, $\beta = 5$, $\omega_1 = 0.003$, $\omega_2 = 0.05$, $\lambda = 0.05$, MaxIter $= 150$ |
| ASSTV | $\lambda_1 = 10$, $\lambda_2 = 60$, $\lambda_3 = 30$, $\gamma = 0.5$, $\alpha = 8$, $\delta = 3$, Maxiter $= 150$ |
| VSNR | Eps $= 0$, $p = 2$, alpha $= 4e-4$, maxit $= 1000$, prec $= 2e-3$, C $= 1000$ |
| SILR | $\delta = 0.01$, $\beta = 1e-04$, $\gamma = 0.01$, $\lambda_2 = 0.7$, $\lambda_3 = 0.5$, $\tau = 0.5$, MaxIter $= 150$ |
| L0 | $\lambda = 10$, $\mu = 0.1$, $\beta_1 = 1$, $\beta_2 = 1$, $\beta_3 = 1$, $\beta_4 = 1$, MaxIter $= 150$ |
| LRSUTV | $\alpha_1 = 20$, $\alpha_3 = 20$, $\alpha_4 = 4$, $\gamma = 0.85$, $\tau = 0.5$, MaxIter $= 150$ |

### 5.6.2. Program Run Time

All test procedures are implemented in MATLAB on a desktop personal computer with a 3.4-GHz CPU and 8 GB RAM. From the perspective of the execution time of the program, our method is not optimal, so we should do corresponding optimization in the next work to further improve the execution efficiency of the program. As for our current work scenario, we will give a brief introduction. We mainly extract CFPN through the algorithm proposed in this paper, and then write the extracted results into the embedded system to deduct the CFPN from the camera in real time. In general, CFPN in sCMOS camera will not change much in a few hours. Although we take a little more time to extract CFPN at a time, the result can be used for several hours, so we are not too sensitive to the running time. The running time of each method is shown in Table 6.

**Table 6.** Running time of various methods. Units are seconds.

| Size | WAFT | UTV | ASSTV | VSNR | SILR | L0 | LRSUTV |
|------|------|-----|-------|------|------|-----|--------|
| $512 \times 512$ | 0.1196 | 4.9194 | 18.0126 | 9.6413 | 20.2170 | 27.9707 | 20.4003 |

## 6. Summary

Although some scholars also use the low-rank sparse method to remove stripes, most of them remove noise from the image itself, and little attention is paid to the structure of the stripes themselves.

Of course, a small number of scholars have considered the structural characteristics of the stripes, but they lack the consideration of the statistical characteristics of the stripes. The stripe removal method we used considers not only the continuity of the image itself, but also the structural and statistical characteristics of the stripes. In terms of parameter adjustment, LRSUTV uses an adaptive scheme for some key parameters, which can automatically adjust the relevant regularization coefficients according to the noise level. This simplifies the adjustment of parameters to some extent. Of course, there are also obvious shortcomings in LRSUTV. First, our method is invalid for tilt stripes. The removal of oblique stripes is occasionally encountered in our work. Secondly, LRSUTV is not currently optimized for parallel computing, so there are inefficient computations. In the next work, we will do some research on the removal of tilted stripes and the improvement of computational efficiency.

**Supplementary Materials:** The matlab codes are available online at http://www.mdpi.com/2076-3417/10/11/3694/s1.

**Author Contributions:** Below is a brief introduction to the relevant contributions of the authors to this paper. The author, T.Z., completed the establishment of the model and the related work of model testing. The author, X.L., mainly works on Algorithm Analysis in this paper. The author, J.L., is mainly engaged in image analysis. The author, Z.X., mainly focuses on the acquisition of experimental data and the analysis of experimental data. The work of each of the above authors has played a key role in the successful completion of the article. All authors have read and agree to the published version of the manuscript.

**Funding:** "National Natural Science Foundation of China", Grant No. 11573066; 11873091 "Yunnan Province Basic Research Plan", Grant No. 2019FA001.

**Conflicts of Interest:** All authors declare no conflict of interest.

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
