# Peer review of "CMOS Fixed Pattern Noise Removal Based on Low Rank Sparse Variational Method"

_applsci, doi:10.3390/app10113694_

Round 1

Reviewer 1 Report

Authors have proposed a novel system for FPN correction and the system provides promising results. Although, Authors have provided sufficient results. I would recommend the authors to look into the field of super resolution and refer the following papers and on how it could be improved further. How they can apply such methodologies for their research. If they could at least provide an insight on how they could use it. It would be great.

[1] https://link.springer.com/article/10.1186/1687-6180-2014-55

Reviewer 2 Report

The manuscript proposes a method for removing Fixed Pattern Noise from images.

The method is based on a variational approach with multiple penalization terms based on different norms. Experiments on synthetic and real data are performed and comparison with several competitor methods is made.

The results provided by the authors are good, however the mathematical formulation of the problem is completely wrong, so that it is difficult to believe that it could have been implemented in an algorithm and give those results. Therefore the manuscript cannot be accepted for publication. Some hints for an eventual resubmission are listed, but authors would be requested to submit the exact codes that were implemented to run all the experiments.

The formulation of the mathematical problem is completely wrong, even though formulation(5) is correct. First of all authors should mention that their method is not fully original, but the norms were also used in other papers relying on variational methods, citing them. In addition they should also be aware that there is a wide statistical literature covering such methods under the terminology of penalization or regularization, where use of L2 and L1 norms has a very precise meaning and role on the solution. Moreover, the formulation in the form min(A) s.t. min(B) is wrong. Two different correct interpretations are a) the minimization of A subject to the constraint that B=alpha, with alpha being some unknown parameter, that gives back the Lagrangian equation as in the manuscript; b) in the statistical framework penalization of the minimization of A by B by means of a trade-off regularization parameter lambda that gives back the same Lagrangian equation. Moreover, in Eq. (6), once, for example, authors set H=dU/dx, the equation H-dU/dx makes no sense because it is 0 be definition. On the other side H is clearly a function (the same applies to J), whereas Eq. (5) is correctly expressed as a form of functional (that is the result of each term is a scalar). There is no way that in Eq. (7) terms with lambda2, lambda3 and lambda4 can be used in the same equation with all other terms, because they give rise to a function, and all other terms to a scalar. Moreover all coefficients alpha, lambda and omega are never defined; it is guessed that they are scalar, therefore the derivative of lambda2 (or lambda3 or lambda4) with respect to x (or y) does not make any sense. Again in the derivation of Eqs. (10) or at line 228 norms of originating equations disappear; in Equation at line 238 from Eq. (10) the term with tau has disappeared. Other inconsistencies are present not listed here.

The authors should also address the following items:

  • English language. The first part of the paper is very poorly written (especially Section up to 4), below any acceptable standard.
  • Authors should give details on the numerical approximation of the equation to be minimized, in particular on the approximation of derivatives
  • Authors should give more details on the choice of the penalization parameters (in a Lagrangian framework they would be estimated during the minimization; in a statistical framework methods like Cross Validation are resorted). This is also important because the authors compare their method with optimized parameters with competitors with rule-of-the-thumb parameters.
  • Variational methods are very time consuming, so that authors should give details on the computational time required by all methods, especially in the case real time applications are needed
  • Figure 3 is not visible properly (and also incomplete)
  • Authors should include among competitors also  VSNR - Variational Stationary Noise Remover by J. Fehrenbach
  • Comparison with wavelet method in the case of solar image is not fair. The double image prevents the method to work properly, as also reported by the authors. However a simple workaround us to analyze the two subimages separately and then to merge them

Reviewer 3 Report

The manuscript deals with source noise pattern removal using low rank techniques. The manuscript has been prepared in rush to a degree that the mathematical analysis around pp. 8 and 9 to be unreadable.

For example Eq. (3) is not formulated correctly. The constraints of an optimization problem cannot be optimization problems.The same applies for Eq. (6).

While Eq. (3) has problems, Eq. (4) is correct.

Eq. (7) is not rendered well.

ADMM is a well founded optimization method. While Eq. (18) is intuitively correct, the pure mathematical typesetting cannot enable this reviewer to follow the exact proof.

Lines 226-233: Please explain why a Fourier transform is needed and where it is applied. Explain what A and B are.

The discussion up to p. 6 can be reduced.

Assess whether improvements in PRSN are statistically significant.

All figures and plots should be discussed in detail.

Overall the manuscript is poorly written. From the very beginning there are instances where punctuation marks are not followed by a blank space.

Reviewer 4 Report

In the reviewed paper, the authors proposed a novel scene-based fixed pattern noise correction method. In general, the paper is in good condition but some things must be improved before the next round of review. My main concerns are:
2) The abstract should have some proper justification of your idea and obtained results. What is the novelty?
3) Used bibliography should be updated - please cite the newest papers from the last 2-4 years top!
4) Used operator * (a star) stands for what? If for multiplication, please use \cdot comment.
5) A related works section is needed.
6) The figures are too small.
7) All captions should be ended with a dot. After writing Figure X should be also a dot.
8) When you write a mathematical formula in the text, please use a $$ environment, for instance in lines 93, 96, etc.
9) Used fonts in the figures are too small.
10) Each formula must be ended with some interruption like a coma.
11) After an equation, when you use the word 'where' there should not be a tab space and the word 'where' should be written with a small letter.
12) When you cite an equation there should be Eq. \eqref{labelOfYourEquation}.
13) Some formulas are formatted very bad like Eq. (11).
14) When your formula is put into a brace like (), and the mathematical content is too big, you should use \left( ... \right)
15) Mathematical formatting must be improved.
16) More experiments on others, more complex datasets must be added.
17) What about theoretical analysis of time/computational analysis?
18) More comparison with the newest (from last 3-4 years) techniques must be added.
19) Add some more results from a more complex dataset - do not use Lana image for that.
20) What about color images?
21) Statistical analysis must be extended.
22) Charts are too small in experimental sections.
23) Summary in section title should be written with a large letter.
24) Conclusions are useless. What is the novelty? What is the impact on other techniques and the current state of knowledge?
25) Why funding information are written in another font?

Round 2

Reviewer 2 Report

The manuscript has fixed some of the flaws present in the first version, admitted by the authors, so that now it can be considered for a review. Its form is still far from getting an acceptance, however goodness of the results deserves one more chance to get the manuscript published

I'm asking the authors to address the following items:

  • p. 6, item 1. Analysis of the gradient is based on the image of Fig. 7. It seems that it is a real image. Therefore how did authors estimate U and S shown in the same image? If they use a method where the vertical gradient of S is null (or approximately), of course this means that the retrieved distributions Y and U will have this property. In other words the comparison of Fig. 8 would show how the method estimates S with a null gradient, not if the stripes themselves have indeed a null gradient. On the other side the property mentioned by the authors seems trivial because they are essentially assuming that stripes coming from CMOS are vertical, therefore their vertical gradient is null and consequently the same approximately hold for Y (apart of noise N).
  • p. 6, item 1. Authors should not use words distribution and histogram at the same time. One shows or the (estimated) distribution, or the histogram; “histogram probability distribution” does not make sense
  • p. 7, item 3. Why does a Gaussian distribution imply an L2 norm?
  • p. 8, eq. 6. Authors mix a continuous formulation (all variables Y, U, S, … are continuous and indeed partial derivatives are correctly invoked) with a discrete formulation involving R2, R3, R4 (which are matrices). They should fix this. By the way, R2, R3, R4 are considered continuous later on (first formula on p. 9).
  • p. 10, Eq. (13-14). Authors explicitly solve Equation (13). How? As far as I know, its prototype is min_u || a-u ||_2 + r ||u||_1 that has an explicit solution as the Soft Thresholding that does not seem to be the one written by authors. On the other side the notation of the authors is fancy: how can one condition the computation of the solution (H) to a condition involving the solution itself (H), that is if it is <0, =0 or >0? The same applies to Eqs. 15-18.
  • p. 11, l. -1. Authors compare their method (with optimized parameters) with competitors that also have tuning parameters. The generally claim to have used default parameters for other methods, but not all of them have default parameters (e.g., VSNR). What exactly authors did? Maybe a Table could help. On the other side on p. 23 Section 5.6.1, authors do not clarify what are the exact values of all the penalization parameters. For example they claim that gamma ranges in [0.65,0.85] when sigma ranges in [5:20]. However value of sigma is an information that is not known in real cases and authors cannot rely on it for comparisons. Simular argument holds for tau (experience based). The same holds for alpha_3 and alpha_4, they are not values but ranges.
  • p. 24, Section 5.6.2: Please number the table. Moreover which units are they? Milliseconds? Seconds? Minutes? Hours? Years?

There are several elementary text and English mistakes spread in the text. The attached file highlights some of them in yellow color.

  • Please pay attention to the editing and to the Journal style (white spaces, Figures sometimes in bold, sometimes not, etc.)
  • lease correct all phrases that do not have an affirmation (e.g., on p. 13, “Because it is visually difficult to distinguish the differences between VSNR, SILR, l0 sparse, LSRUTV methods.” that makes no sense).
  • Please avoid any personal verb in the manuscript (e.g., on p. 13 it is difficult “for me”…). On the other side there are 4 authors in the manuscript!

Finally, the source code with data has to be mandatorily attached in an eventual next resubmission

Reviewer 3 Report

I went through the revised manuscript. Its quality has been improved. However, in p.9 there is room for further improvement.

Please apply the Fourier transform to the constraint of the optimization problem and derive A and B. The left hand side of the equation should be the Fourier Transform of the operator B applied to U, i.e.:

\mathcal{F}\{ BU\}=

\mathcal{F} \{ (a_1 + \omega_2 \frac{\partial^2}{\partial x^2} +

\omega_3 \frac{\partial^2}{\partial y^2} + \omega_4 \frac{\partial^2}{\partial y^2} )U \} = 

\mathcal{F} \{ a_1 U + \omega_2 \frac{\partial^2 U}{\partial x^2} +

\omega_3 \frac{ \partial^2 U}{\partial y^2} + \omega_4 \frac{\partial^2 U}{\partial y^2} \}

Proceed similarly to the derivation of A.

The time unit (sec?, msec?) is not explicitly defined in Section 5.6.2.

The references do not adhere to a common style.

Reviewer 4 Report

If a scientific article is not a review, it should have some novelty in relation to existing methods. This is not always a new technique, but modification or improvement. So my previous remark was relevant. In the current version, the authors wrote what exactly is introduced by the article and it can be accepted.

Round 3

Reviewer 2 Report

The authors did not address properly the items raised in my latest request. In addition the formula they revised contains one more new mistake.

Nevertheless, the methodology developed by authors is interesting and the results shown good. Therefore I am keen to give the authors a (last) chance to revised the manuscript without mistakes and addressing the items I raised and that were not satisfactorily addressed. In summary I will accept the manuscript if there are not mistakes and the procedure is described without any ambiguity.

Items to be addressed:

  • new formula (16) contains a new mistake in the middle formula. Please correct it and align the 3 formulas.
  • concerning my issue on discrete/continuous operators, a proper formulation cannot mix discrete and continuous operators, which is still done in the new formulation. One a) or entirely writes the equation in continuous form and then at the end specifies how approximates the continuous operators with discrete ones (e.g., derivatives), or b) writes the equations directly in discrete form. The formulation of authors is hybrid: they clearly use a continuous notation (otherwise they could not obtain formulas in lines 277-287; otherwise they would not need to specify at line 340-343 that operators have been discretized), however at lines 253-266 they still deal with discrete R, completely mixing the two notations. 
  • Also at line 249 s.t. has to be removed
  • Issue on the tuning parameters. To give a range for parameters does not mean to fix parameters. If one wants to use a method, he/she must know which value to assign to parameters. In addition, if a method is compared with competitors for which, as mentioned in the manuscript, default (therefore fixed) values have been used, then for a fair comparison a fixed value of parameters (not a range) has to be provided. To give a range can be certainly tolerated, mentioning that exact values of the parameters have to be manually tuned with a particular image (as the remark of the authors suggest), provided that a similar optimization of parameters is performed also with competitor methods.
